# Distributed Adversarial Training to Robustify Deep Neural Networks at Scale

## Abstract

Current deep neural networks are vulnerable to adversarial attacks, where adversarial perturbations to the inputs can change or manipulate classification. To defend against such attacks, an effective and popular approach, known as *adversarial training*, has been shown to mitigate the negative impact of adversarial attacks by virtue of a min-max robust training method. While effective, this approach is difficult to scale well to large models on large datasets (e.g., ImageNet) in general. To address this challenge, we propose *distributed adversarial training (DAT)*, a *large-batch* adversarial training framework implemented over multiple machines. DAT supports one-shot and iterative attack generation methods, gradient quantization, and training over labeled and unlabeled data. Theoretically, we provide, under standard conditions in the optimization theory, the convergence rate of DAT to the first-order stationary points in general non-convex settings. Empirically, on ResNet-18 and -50 under CIFAR-10 and ImageNet, we demonstrate that DAT either matches or outperforms state-of-the-art robust accuracies and achieves a graceful training speedup.

## 1 Introduction

The rapid increase of research in deep neural networks (DNNs) and their adoption in practice is, in part, owed to the significant breakthroughs made with DNNs in computer vision (Alom et al., 2018). Yet, with the apparent power of DNNs, there remains a serious weakness of robustness. That is, DNNs can easily be manipulated (by an adversary) to output drastically different classifications and can be done so in a controlled and directed way. This process is known as an adversarial attack and considered as one of the major hurdles in using DNNs in security critical and real-world applications (Goodfellow et al., 2015; Szegedy et al., 2013; Carlini & Wagner, 2017; Papernot et al., 2016; Kurakin et al., 2016; Eykholt et al., 2018; Xu et al., 2019b).

Methods to train DNNs being robust against adversarial attacks are now a major focus in research (Xu et al., 2019a). But most are far from satisfactory (Athalye et al., 2018) with the exception of the adversarial training (AT) approach (Madry et al., 2017b). AT is a min-max robust training method that minimizes the worst-case training loss at adversarially perturbed examples. AT has inspired a wide range of state-of-the-art defenses (Kannan et al., 2018; Ross & Doshi-Velez, 2018; Moosavi-Dezfooli et al., 2019; Zhang et al., 2019b; Wang et al., 2019b; Sinha et al., 2018; Chen et al., 2019; Boopathy et al., 2020; Wong & Kolter, 2017; Dvijotham et al., 2018; Stanforth et al., 2019; Carmon et al., 2019; Shafahi et al., 2019; Zhang et al., 2019a; Wong et al., 2020), which ultimately resort to min-max optimization. However, these methods, together with AT, are generally difficult to scale well to large networks on large datasets.

While scaling AT is important, doing so effectively is non-trivial. We find that scaling AT with the direct solution of *distributing* the data batch across *multiple* machines may not work and leaves many unanswered questions. First, if the direct solution does not allow for scaling batch size with machines, then it does not speed the process up and leads to a significant amount of communication costs (considering that the number of training iterations is not reduced over a fixed number of epochs). Second, without proper design, the direct application of a large batch size to distributed adversarial training introduces a significant loss in both normal accuracy and adversarial robustness (e.g., more than $10\%$ performance drops for ResNet-18 on CIFAR-10 shown by our experiments). Third, the

direct approach does not confer a general algorithmic framework, which is needed in order to support different variants of AT, large-batch optimization, and efficient communication.

Taking all factors into consideration, a question that naturally arises is: *Can we speed up AT by leveraging distributed learning with full utility of multiple computing nodes (machines), even when each only has access to limited GPU resources?* Although a few works made empirical efforts to scale AT up by simply using multiple computing nodes (Xie et al., 2019; Kang et al., 2019; Qin et al., 2019), they were limited to specific use cases and lacked a thorough study on when and how distributed learning helps, either in theory or in practice. By contrast, we propose a principled and theoretically-grounded *distributed (large-batch) adversarial training* (DAT) framework by making full use of the computing capability of multiple data-locality (distributed) machines, and show that DAT expands the capacity of data storage and the computational scalability. We summarize our main contributions as below.

**Contributions**   *(i)* We provide a general problem formulation of DAT, which supports multiple distributed variants of AT, e.g., supervised AT and semi-supervised AT. *(ii)* We propose a principled algorithmic framework for DAT, which, different from conventional AT, supports large-batch DNN training (without losing performance over a fixed number of epochs) and allows the transmission of compressed gradients for efficient communication. *(iii)* We theoretically quantify the convergence speed of DAT to the first-order stationary points in general non-convex settings at a rate of $O(1/\sqrt{T})$, where $T$ is the total number of iterations. This result matches the standard convergence rate of classic training algorithms, e.g., stochastic gradient descent (SGD), for only the minimization problems. *(iv)* We make a comprehensive empirical study on DAT, showing that it not only speeds up in training large models on large datasets but also matches (and even exceeds) state-of-the-art robust accuracies in different attacking and learning scenarios. For example, DAT on ImageNet with $6 \times 6$ (machines $\times$ GPUs per machine) yields $38.45\%$ robust accuracy (comparable to $40.38\%$ from AT) but only requires 16.3 hours training time (6 times larger batch size allowed in DAT), exhibiting 3.1 times faster than AT on a single machine of 6 GPUs. We also make a significant effort to evaluate the empirical performance of DAT across different computing configurations at different learning regimes, e.g., semi-supervised learning and transfer learning.

## 2   RELATED WORK

**Training robust classifiers**   The lack of robustness of DNNs has promoted a rapid expansion of defenses against adversarial attacks, ranging from heuristic defenses to robust (min-max) optimization based approaches. However, many heuristic strategies are easily bypassed by stronger adversaries due to the presence of obfuscated gradients (Athalye et al., 2018). By contrast, the min-max optimization-based training methods are generally able to offer significant gains in robustness. AT (Madry et al., 2017b), the first known min-max optimization-based defense, has inspired a wide range of other effective defenses. Examples include adversarial logit pairing (Kannan et al., 2018), input gradient or curvature regularization (Ross & Doshi-Velez, 2018; Moosavi-Dezfooli et al., 2019), trade-off between robustness and accuracy (TRADES) (Zhang et al., 2019b), distributionally robust training (Sinha et al., 2018), dynamic adversarial training (Wang et al., 2019b), robust input attribution regularization (Chen et al., 2019; Boopathy et al., 2020), certifiably robust training (Wong & Kolter, 2017; Dvijotham et al., 2018), and semi-supervised robust training (Stanforth et al., 2019; Carmon et al., 2019).

In particular, some recent works proposed *fast but approximate* AT algorithms, such as 'free' AT (Shafahi et al., 2019), you only propagate once (YOPO) (Zhang et al., 2019a), and fast gradient sign method (FGSM) based AT (Wong et al., 2020). These algorithms achieve speedup in training by simplifying the inner maximization step of AT. Although there is vast literature on min-max optimization based robust training, it is designed for centralized model training and without care about the scalability issue in AT. In (Xie et al., 2019), although a distributed version of vanilla AT was implemented via the large-batch SGD algorithm (Goyal et al., 2017), it is significantly different from our proposal. Compared to (Xie et al., 2019), we adopt a different distributed training recipe (layer-wise adaptive learning rate method vs. SGD), and make in-depth theoretical analysis of quantifying the convergence rate of DAT.

**Distributed model training**   Distributed optimization has been found to be effective for the standard training of machine learning models (Dean et al., 2012; Goyal et al., 2017; You et al., 2019; Chen et al., 2020). In contrast to centralized optimization, distributed learning enables increasing the batch size proportional to the number of computing nodes/machines. However, it is challenging to train a

model via large-batch optimization without incurring accuracy loss compared to the standard training with same number of epochs (Krizhevsky, 2014; Keskar et al., 2016). To tackle this challenge, it was shown in (You et al., 2017; 2018; 2019) that adaptation of learning rates to the increase of the batch size is an essential mean to boost the performance of large-batch optimization. A layer-wise adaptive learning rate strategy was then proposed to speed up the training as well as preserve the accuracy. Although these works have witnessed several successful applications of distributed learning in training *standard* image classifiers, they leave the question of how to build *robust* DNNs with DAT open. In this paper, we show that the power of layer-wise adaptive learning rate also applies to DAT. Since distributed learning introduces machine-machine communication overhead, another line of work (Alistarh et al., 2017; Yu et al., 2019; Bernstein et al., 2018; Wangni et al., 2018; Stich et al., 2018; Wang et al., 2019a) focused on the design of communication-efficient distributed optimization algorithms.

The study on distributed learning is extensive, but the problem of distributed min-max optimization is less explored, with some exceptions (Srivastava et al., 2011; Notarnicola et al., 2018; Hanada et al., 2017; Tsaknakis et al., 2020; Liu et al., 2019a;b). A key difference to our work is that none of the aforementioned literature studied the *large-batch* min-max optimization with its applications to training *robust* DNNs, neither theoretically nor empirically. While there are recent proposed algorithms for training Generative Adversarial Nets (GANs) (Liu et al., 2019a;b), training robust DNNs against adversarial examples is intrinsically different from GAN training. In particular, training robust DNNs requires inner maximization with respect to each training data rather than empirical maximization with respect to model parameters. Such an essential difference leads to different optimization goals, algorithms, convergence analyses and implementations.

## 3 PROBLEM FORMULATION

In this section, we first review the standard setup of adversarial training (AT) (Madry et al., 2017b). We then motivate the need of distributed AT (DAT) and propose a general min-max setup for DAT.

**Adversarial training**   AT (Madry et al., 2017b) is a min-max optimization method for training robust ML/DL models against adversarial examples (Goodfellow et al., 2015). Formally, AT solves the problem

$$\underset{\boldsymbol{\theta}}{\text{minimize}}\ \mathbb{E}_{(\mathbf{x},y)\in\mathcal{D}}\left[\underset{\|\boldsymbol{\delta}\|_\infty\leq\epsilon}{\text{maximize}}\ \ell(\boldsymbol{\theta},\mathbf{x}+\boldsymbol{\delta};y)\right], \tag{1}$$

where $\boldsymbol{\theta}\in\mathbb{R}^d$ denotes the vector of model parameters, $\boldsymbol{\delta}\in\mathbb{R}^n$ is the vector of input perturbations within an $\ell_\infty$ ball of the given radius $\epsilon$, namely, $\|\boldsymbol{\delta}\|_\infty\leq\epsilon$, $(\mathbf{x},y)\in\mathcal{D}$ corresponds to the training example $\mathbf{x}$ with label $y$ in the dataset $\mathcal{D}$, and $\ell$ represents a pre-defined training loss, e.g., the cross-entropy loss. The rationale behind problem (1) is that the model $\boldsymbol{\theta}$ is robustly trained against the *worst-case* loss induced by the adversarially perturbed samples. It is worth noting that the AT problem (1) is *different* from conventional stochastic min-max optimization problems (e.g., GANs training (Goodfellow et al., 2014)). Note that in (1), the stochastic sampling corresponding to the expectation over $(\mathbf{x},\mathbf{y})\in\mathcal{D}$ is conducted *prior to* the inner maximization operation. Such a difference leads to the *sample-specific* adversarial perturbation $\boldsymbol{\delta}(\mathbf{x}):=\text{maximize}_{\|\boldsymbol{\delta}\|_\infty\leq\epsilon}\ \ell(\boldsymbol{\theta},\mathbf{x}+\boldsymbol{\delta};y)$.

**Distributed AT (DAT)**   The need of AT in a *distributed* setting arises from at least the following two aspects: 1) Training data are distributed, provided by multiple parties, which expands the individual capability of data storage or data privacy. 2) Computing units are often distributed, provided by distributed machines, which enables large-batch optimization and thus improves the AT's scalability.

Let us consider a popular parameter-server model of distributed learning (Dean et al., 2012). Formally, there exist $M$ workers each of which has access to a local dataset $\mathcal{D}^{(i)}$, and thus $\mathcal{D}=\cup_{i=1}^M\mathcal{D}^{(i)}$. There also exists a server/master node (e.g., one of workers could perform as server), which collects local information (e.g., individual gradients) from the other workers to update the model parameters $\boldsymbol{\theta}$. Spurred by (1), DAT solves problems of the following generic form,

$$\underset{\boldsymbol{\theta}}{\text{minimize}}\ \frac{1}{M}\sum_{i=1}^M\underbrace{\left\{\lambda\mathbb{E}_{(\mathbf{x},y)\in\mathcal{D}^{(i)}}\left[\ell(\boldsymbol{\theta};\mathbf{x},y)\right]+\mathbb{E}_{(\mathbf{x},y)\in\mathcal{D}^{(i)}}\left[\underset{\|\boldsymbol{\delta}\|_\infty\leq\epsilon}{\text{maximize}}\ \phi(\boldsymbol{\theta},\boldsymbol{\delta};\mathbf{x},y)\right]\right\}}_{=:\ f_i(\boldsymbol{\theta};\mathcal{D}^{(i)})}, \tag{2}$$

where $f_i$ denotes the local cost function at the $i$th worker, $\phi$ is a robustness regularizer against the input perturbation $\boldsymbol{\delta}$, and $\lambda \geq 0$ is a regularization parameter that strikes a balance between the training loss and the worst-case robustness regularization. In (2), if $M = 1$, $\mathcal{D}^{(1)} = \mathcal{D}$, $\lambda = 0$ and $\phi = \ell$, then the DAT problem reduces to the AT problem (1). We cover two categories of (2):

● DAT with labeled data: In (2), we consider $\phi(\boldsymbol{\theta}, \boldsymbol{\delta}; \mathbf{x}, y) = \ell(\boldsymbol{\theta}, \mathbf{x} + \boldsymbol{\delta}; y)$ with labeled training data $(\mathbf{x}, y) \in \mathcal{D}^{(i)}$ for $i \in [M]$. Here $[M]$ denotes the integer set $\{1, 2, \ldots, M\}$.

● DAT with unlabeled data: In (2), different from DAT with labeled data, we have $\mathcal{D}^{(i)}$ with an unlabeled dataset $\mathcal{U}^{(i)}$ (namely, $\mathcal{U}^{(i)} \subseteq \mathcal{D}^{(i)}$), and define the robust regularizer $\phi$ as (Stanforth et al., 2019; Zhang et al., 2019b):

$$\phi(\boldsymbol{\theta}, \boldsymbol{\delta}; \mathbf{x}) = \mathrm{CE}(\mathbf{z}(\mathbf{x} + \boldsymbol{\delta}; \boldsymbol{\theta}), \mathbf{z}(\mathbf{x}; \boldsymbol{\theta})). \tag{3}$$

Here $\mathbf{z}(\mathbf{x}; \boldsymbol{\theta})$ represents the probability distribution over class labels predicted by the model $\boldsymbol{\theta}$, and CE denotes the cross-entropy function.

## 4 A UNIFIED ALGORITHMIC FRAMEWORK FOR DAT

In this section, we will introduce the algorithmic framework of DAT, where we address three main challenges of algorithm design in DAT: a) inner maximization; b) gradient quantization/compression; and c) outer large-batch training by leveraging state-of-the-art adversarial defense and distributed optimization techniques. We also achieve the first theoretical convergence rate result of min-max optimization based robust training in a distributed large-batch setting.

We first discuss the key components of DAT and their respective roles in performing scalable training; see the meta-form of DAT in Algorithm 1 (or detailed Algorithm A1). The proposed DAT contains three algorithmic blocks. In the *first* block (Steps 3-8 of Algorithm A1), every distributed worker calls for a maximization oracle to obtain the adversarial perturbation for each sample within a data batch, then computes the gradient of the local cost function $f_i$ in (2) with respect to (w.r.t.) model parameters $\boldsymbol{\theta}$. And every worker is allowed to quantize/compress the local gradient prior to transmission to a server. In the *second* block (Steps 9-10 of Algorithm A1), the server aggregates the local gradients, and transmits the aggregated gradient (or the quantized gradient) to the other workers. In the *third* block (Steps 11-13 of Algorithm A1), the model parameters are eventually updated by a minimization oracle at each worker based on the received gradient information from the server.

In contrast to standard AT, DAT allows for using a $M$ times larger batch size to update the model parameters $\boldsymbol{\theta}$. Thus, given the same number of epochs, DAT takes $M$ fewer gradient updates than AT. In addition, distributed learning introduces communication overhead. To address this issue, it is optional to perform gradient quantization at both worker and server sides when a very large model is possibly trained. We elaborate on DAT in what follows.

**Algorithm 1** Meta-form of DAT Algorithm A1

---
1: **for** Iteration $t = 1, 2, \ldots, T$
2:   **for** Worker $i = 1, 2, \ldots, M$     ▷ Block 1
3:     Sample-wise attack generation (A1)
4:     Local gradient computation (A2)
5:     Worker-server communication (optional)
6:   **end for**
7: Gradient aggregation at server (A3) ▷ Block 2
8: Server-worker communication (optional)
9: **for** Worker $i = 1, 2, \ldots, M$     ▷ Block 3
10:     Model parameter update (A4)
11: **end for**

---

**Inner maximization: Iterative and one-shot solutions** In DAT, each worker calls for an inner maximization oracle to generate adversarial perturbations (Step 1 of Algorithm 1). We consider two solvers of perturbation generation: iterative projected gradient descent (PGD) method used in standard AT (Madry et al., 2017a) and one-shot fast gradient sign method (FGSM) (Goodfellow et al., 2015). We specify attack generation in the unified form

$$\boldsymbol{\delta}_t^{(i)}(\mathbf{x}) = \mathbf{z}_K, \quad \mathbf{z}_k = \Pi_{[-\epsilon, \epsilon]^d}[\mathbf{z}_{k-1} + \alpha \cdot \mathrm{sign}(\nabla_{\boldsymbol{\delta}} \phi(\boldsymbol{\theta}_t, \mathbf{z}_{k-1}; \mathbf{x}))], \ k \in [K], \tag{4}$$

where $K$ is the total number of iterations in the inner loop, the cases of $K = 1$ and $K > 1$ correspond to iterative PGD attack and FGSM attack respectively, $\mathbf{z}_k$ denotes the PGD update of $\boldsymbol{\delta}$ at the $k$th iteration, $\mathbf{z}_0$ is a given intial point, $\Pi_{[-\epsilon, \epsilon]^d}(\cdot)$ denotes the projection onto the box constraint $[-\epsilon, \epsilon]^d$, $\alpha > 0$ is a given step size, and $\mathrm{sign}(\cdot)$ denotes the element-wise sign operation. The recent work (Wong et al., 2020) showed that if FGSM is conducted with random initialization $\mathbf{z}_0$ and a proper step

size, e.g., $\alpha = 1.25\epsilon$, then FGSM can be as effective as iterative PGD in robust training. Indeed, we will show in Sec. 5 that the effectiveness of our proposed DAT-FGSM algorithm echoes the finding in Wong et al. (2020). We remark that other techniques (Shafahi et al., 2019; Zhang et al., 2019a) can also be used to simplify inner maximization, however, we focus on FGSM since it is the most computationally-light.

**Gradient quantization**    In contrast to standard AT, DAT requires worker-server communications (Steps 5 and 8 of Algorithm 1). That is, if a single-precision floating-point data type is used, then DAT needs to transmit $32d$ bits per worker-server communication at each iteration. Here recall that $d$ is the dimension of $\boldsymbol{\theta}$. In order to reduce the communication cost, DAT has the option to quantize the transmitted gradients using a fixed number of bits fewer than 32. We specify the gradient quantization operation as the *randomized quantizer* (Alistarh et al., 2017; Yu et al., 2019). In Sec. 5 we will show that DAT, combined with gradient quantization, still leads to a competitive performance. For example, the robust accuracy of ResNet-50 trained by a 8-bit DAT (performing quantization at Step 5 of Algorithm 1) for ImageNet is just $0.55\%$ lower than the robust accuracy achieved by the 32-bit DAT. It is also worth mentioning that the All-reduce communication protocol can be regarded as a special case of the parameter-server setting considered in Algorithm 1 when every worker performs as a server. In this case, the communication network becomes fully connected and the server-worker quantization (Step 8 of Algorithm 1) can be mitigated.

**Outer minimization by layerwise adaptive learning rate (LALR)**    In DAT, the aggregated gradient (Step 7 in Algorithm 1) used for updating model parameters (Step 10 in Algorithm 1) is built on the data batch that is $M$ times larger than standard AT. The recent works (You et al., 2019; 2017) showed that the use of LALR is the key to succeed in training standard DNNs with large data batch. Spurred by that, we incorporate LALR in DAT. Specifically, the parameter updating operation $\mathcal{A}$ in Eq. (A4) is given by

$$\boldsymbol{\theta}_{t+1,i} = \boldsymbol{\theta}_{t,i} - \frac{\tau(\|\boldsymbol{\theta}_{t,i}\|_2) \cdot \eta_t}{\|\mathbf{u}_{t,i}\|_2} \cdot \mathbf{u}_{t,i}, \quad \forall i \in [h], \tag{5}$$

where $\boldsymbol{\theta}_{t,i}$ denotes the $i$th-layer parameters, $h$ is the number of layers, $\mathbf{u}_t$ is a descent direction computed based on the first-order information $Q(\hat{\mathbf{g}}_t)$, $\tau(\|\boldsymbol{\theta}_{t,i}\|_2) = \min\{\max\{\|\boldsymbol{\theta}_{t,i}\|_2, c_l\}, c_u\}$ is a *layerwise* scaling factor of the *adaptive* learning rate $\frac{\eta_t}{\|\mathbf{u}_{t,i}\|_2}$, $c_l = 0$ and $c_u = 10$ are set in our experiments (see Appendix 4.2 for results on tuning $c_u$), and $\boldsymbol{\theta}_t = [\boldsymbol{\theta}_{t,1}^\top, \ldots, \boldsymbol{\theta}_{t,h}^\top]^\top$. In (5), the specific form of the descent direction $\mathbf{u}_t$ is determined by the optimizer employed. For example, if the adaptive momentum (Adam) method is used, then $\mathbf{u}_t$ is given by the exponential moving average of past gradients scaled by square root of exponential moving averages of squared past gradients (Reddi et al., 2018; Chen et al., 2018). Such a variant of (5) that uses Adam as the base algorithm is also known as LAMB (You et al., 2019) in standard training. However, it was elusive if the advantage of LALR is preserved in large-batch min-max optimization. We show in both theory and practice that the use of LALR can significantly boost the performance of DAT with large data batch.

CONVERGENCE ANALYSIS OF DAT

To the best of our knowledge, none of existing work tackled the convergence of DAT and took into account LALR and gradient quantization even in standard AT, although AT has been proved with convergence guarantees (Wang et al., 2019b; Gao et al., 2019). DAT needs to quantify the descent errors from multiple sources (namely, gradient estimation, quantization, adaptive learning rate, and inner maximization oracle). In particular, the incorporation of LALR makes our analysis of DAT highly non-trivial. The fundamental challenge lies in the nonlinear coupling between the biased gradient estimate resulted from LALR and the additional error generated from alternating update in AT. In our theoretical results, we show that even in the case where the gradient estimate is a function of the AT variables, the estimate bias resulted from the layer-wise normalization can still be compensated by increasing the batch-size so that the convergence rate of DAT achieves a linear speedup of reducing gradient estimate error w.r.t. the increasing number of computing nodes.

Upon defining $\Psi(\boldsymbol{\theta}) := \frac{1}{M} \sum_{i=1}^M f_i(\boldsymbol{\theta}; \mathcal{D}^{(i)})$ in (2), we measure the convergence of DAT by the first-order stationarity of $\Psi$. Prior to convergence analysis, we impose the following assumptions: (A1) $\Psi(\boldsymbol{\theta})$ is with layer-wise Lipschitz continuous gradients; (A2) $\phi(\boldsymbol{\theta}, \boldsymbol{\delta}; \mathbf{x})$ in Eq. (A1) is strongly concave with respect to $\boldsymbol{\delta}$ and with Lipschitz continuous gradients; (A3) Stochastic gradient estimate

in Eq. (A2) is unbiased and has bounded variance for each worker denoted by $\sigma^2$. Note that the validity of (A2) could be justified from distributional robust optimization (Sinha et al., 2018; Wang et al., 2019b). It is also needed for tractability of analysis. We refer readers to Appendix 2.1 for more justifications on our assumptions (A1)-(A3). In Theorem 1, we present the sub-linear rate of DAT.

**Theorem 1.** *Suppose that assumptions A1-A3 hold, the inner maximizer Eq. (A1) provides a $\varepsilon$-approximate solution (i.e., the $\ell_2$-norm of inner gradient is upper bounded by $\varepsilon$), and the learning rate is set by $\eta_t \sim \mathcal{O}(1/\sqrt{T})$, then $\{\boldsymbol{\theta}_t\}_{t=1}^T$ generated by DAT yields the following convergence rate*

$$\frac{1}{T} \sum_{t=1}^T \mathbb{E}\|\nabla_{\boldsymbol{\theta}} \Psi(\boldsymbol{\theta}_t)\|_2^2 = \mathcal{O}\left(\frac{1}{\sqrt{T}} + \frac{\sigma}{\sqrt{MB}} + \min\left\{\frac{d}{4^b}, \frac{\sqrt{d}}{2^b}\right\} + \varepsilon\right), \tag{6}$$

*where $b$ denotes the number of quantization bits, and $B = \min\{|\mathcal{B}_t^{(i)}|, \forall t, i\}$ stands for the smallest batch size per worker.*

**Proof**: Please see Appendix 3. $\qquad\qquad\qquad\qquad\qquad\qquad\qquad\qquad\qquad\qquad\qquad\square$

The error rate given by (6) involves four terms. The term $\mathcal{O}(1/\sqrt{MB})$ characterizes the benefit of using the large per-worker batch size $B$ and $M$ computing nodes in DAT. It is introduced since the variance of adaptive gradients (i.e., $\sigma^2$) is reduced by a factor $1/MB$, where $1/M$ corresponds to the linear speedup by $M$ machines. In (6), the term $\min\{\frac{d}{4^b}, \frac{\sqrt{d}}{2^b}\}$ arises due to the variance of compressed gradients, and the other two terms imply the dependence on the number of iterations $T$ as well as the $\varepsilon$-accuracy of the inner maximization oracle. We highlight that our convergence analysis (Theorem 1) is not barely a combination of LALR-enabled standard training analysis (You et al., 2019; 2017) and adversarial training convergence analysis (Wang et al., 2019b; Gao et al., 2019). Different from the previous work, we address the fundamental challenges in (a) quantifying the descent property of the objective value at the presence of multi-source errors during alternating min-max optimization, and (b) deriving the theoretical relationship between large data batch (across distributed machines) and the eventual convergence error of DAT.

## 5 EXPERIMENTS: DAT FOR ROBUST IMAGE CLASSIFICATION

In this section, we empirically evaluate DAT and show its success in training robust deep neural networks (DNNs) over CIFAR and ImageNet datasets. We measure the performance of DAT in the following four aspects: a) accuracies against clean and adversarial test inputs, b) scalability to multiple computing nodes, c) incorporation of unlabeled data, and d) transferability of pre-trained model by DAT.

**DNN models and Datasets**   We use the DNN models Pre-act ResNet-18 (He et al., 2016b) and ResNet-50 (He et al., 2016a) for image classification, where the former is shortened as ResNet-18. We train these models under datasets CIFAR-10 and ImageNet, but preserve ResNet-18 for CIFAR-10 only. We also acquire unlabeled data from 80 Million Tiny Images following (Carmon et al., 2019). When studying pre-trained model's transferability, CIFAR-100 is used as a target dataset for down-stream classification.

**Computing resources**   We train a DNN using $p$ computing nodes, each of which contains $q$ GPUs (Nvidia V100 or P100). Nodes are connected with 1Gbps ethernet. *A configuration of computing resources is noted by $p \times q$.* If $p > 1$, then the training is conducted in a *distributed* manner. And we split training data into $p$ subsets, each of which is stored at a local node. Based on our resources, in the CIFAR experiments, we consider $p \in \{1, 6, 18, 24\}$ machines, each of which has 1 GPU. In the ImageNet experiments, we consider $p \in \{1, 6\}$ machines, each of which has 6 GPUs. Based on our computing resource budget, we are not able to use as many GPUs as (Xie et al., 2019; Kannan et al., 2018). However, as will be evident later, our results clearly demonstrate the advantages of our proposal in applicability and scalability across various computing configurations and adversarial scenarios. In particular, our used batch size $6 \times 512 = 3072$ on ImageNet has only access to 36 GPUs. By contrast, Xie et al. (2019) used the batch size 4096 across 128 GPUs.

**Training setting**   We consider 2 *variants* of DAT: 1) *DAT-PGD*, namely, Algorithm A1 with applying (iterative) PGD as the inner maximization oracle; and 2) *DAT-FGSM*, namely, Algorithm A1 with

use of FGSM as the inner maximization oracle. Additionally, we consider $4$ training *baselines*: 1) *AT* (Madry et al., 2017b); 2) *Fast AT* (Wong et al., 2020); 3) *DAT w/o LALR*, namely, a direct distributed implementation of AT, which is in the form of DAT-PGD or DAT-FGSM but *without* considering LALR; and 4) *DAT-LSGD* (Xie et al., 2019), namely, a distributed implementation of large-batch SGD (LSGD) for standard AT. We remark that both AT and Fast AT are centralized training methods. In our setup, the number of GPUs is limited to 6 at a single machine, and thus the largest batch size that the centralized method can use is around 2048 for CIFAR-10 and 85 for ImageNet in our case. We also find that the direct implementation of Fast-AT in a distributed way leads to a quite poor scalability versus the growth of batch size, and thus a worse distributed baseline than DAT-FGSM w/o LALR. Lastly, we remark that the work (Xie et al., 2019) proposed modifying a model architecture by incorporating feature denoising. In contrast, DAT does not call for architecture modification. Thus, to enable a fair comparison, we use the same training recipe LSGD as (Xie et al., 2019) in the DAT setting, leading to the considered distributed training baseline DAT-LSGD.

Unless specified otherwise, we choose the training perturbation size $\epsilon = 8/255$ and $2/255$ for CIFAR and ImageNet respectively, where recall that $\epsilon$ was defined in (1). We also choose 10 steps and 4 steps for PGD attack generation in DAT (and its variants) under CIFAR and ImageNet, respectively. Such training settings are consistent with previous state-of-the-art (Zhang et al., 2019a; Wong et al., 2020). The number of training epochs is given by 100 for CIFAR-10 and 30 for ImageNet. Note that the adversarially robust deep learning could be sensitive to the step size (learning rate) choice (Rice et al., 2020). For example, the use of a cyclic learning rate trick can further accelerate the Fast AT algorithm (Wong et al., 2020). However, such a trick becomes less effective when the batch size becomes larger (namely, the number of iterations gets smaller); see Appendix 4.1. Meanwhile, the sensitivity of adversarially model training to step size can be mitigated by using early-stop remedy due to the existence of robust overfitting (Rice et al., 2020). Spurred by that, we use the standard piecewise decay step size and an early-stop strategy during robust training. We refer readers to Appendix 4.2 for more implementation details.

**Evaluation setting** For adversarial evaluation, we report robust test accuracy (RA) of a learned model against PGD attacks (Madry et al., 2017b) and C&W attack (Carlini & Wagner, 2017). Unless specified otherwise, we choose the perturbation size same as the training $\epsilon$ in evaluation, and the number of PGD steps is selected as 20 and 10 for CIFAR and ImageNet, respectively. In addition to RA, we also measure the standard test accuracy (TA) of a model against normal examples. All experiments are run $3$ times with different random seeds, and the mean metrics are reported[1].

Table 1: Overall performance of DAT (in gray color), compared with baselines, in TA (%), RA (%), communication time per epoch (seconds), and total training time (including communication time) per epoch (seconds). For brevity, '$p \times q$' represents '# nodes $\times$ # GPUs per node', 'Comm.' represents communication cost, and 'Tr. Time' represents training time.

| Method | CIFAR-10, ResNet-18 | | | | | |
|---|---|---|---|---|---|---|
| | $p \times q$ | Batch size | TA (%) | RA (%) | Comm. | Tr. time |
| AT | $1 \times 1$ | 2048 | 82.94 | 38.54 | NA | 218 |
| Fast AT | $1 \times 1$ | 2048 | 81.58 | 38.34 | NA | 52 |
| DAT-PGD w/o LALR | $18 \times 1$ | $18 \times 2048$ | 55.59 | 26.83 | 3.4 | 22 |
| DAT-FGSM w/o LALR | $18 \times 1$ | $18 \times 2048$ | 52.35 | 28.90 | 3.1 | 8 |
| DAT-LSGD | $18 \times 1$ | $18 \times 2048$ | 64.15 | 34.12 | 3.2 | 22 |
| DAT-PGD | $18 \times 1$ | $18 \times 2048$ | 80.28 | 38.44 | 3.4 | 22 |
| DAT-FGSM | $18 \times 1$ | $18 \times 2048$ | 73.42 | 38.55 | 3.1 | 8 |
| DAT-FGSM | $24 \times 1$ | $24 \times 2048$ | 72.76 | 39.82 | 2.0 | 5 |
| | ImageNet, ResNet-50 | | | | | |
| AT | $1 \times 6$ | 512 | 62.70 | 40.38 | NA | 6022 |
| Fast AT | $1 \times 6$ | 512 | 58.99 | 40.78 | NA | 1544 |
| DAT-PGD w/o LALR | $6 \times 6$ | $6 \times 512$ | 57.09 | 34.02 | 865 | 1932 |
| DAT-FGSM w/o LALR | $6 \times 6$ | $6 \times 512$ | 55.04 | 35.03 | 863 | 1080 |
| DAT-PGD | $6 \times 6$ | $6 \times 512$ | 63.75 | 38.45 | 898 | 1960 |
| DAT-FGSM | $6 \times 6$ | $6 \times 512$ | 58.32 | 41.48 | 859 | 1109 |

In our experiments, we consider three different communication protocols, Ring-AllReduce (with one-sided quantization), parameter-server (with double quantization), and high performance computing (HPC) setting (without quantization). To measure the communication time, We use TORCH.DISTRIBUTED package with gloo and nccl as communication backend[2]. We then measure the time of required worker-server communications per epoch. We use a time module to measure communication time with TORCH.DISTRIBUTED.BARRIER to synchronize all processes on each node.

---

[1] Code will be released
[2] https://pytorch.org/docs/stable/distributed.html

**Overall performance of DAT** In Table 1, we evaluate our proposals and baselines in TA, RA, communication and computation efficiency. Note that AT and Fast AT are centralized training methods in single node under the same number of epochs as distributed training.

We observe that the direct extension from AT to DAT (namely, DAT-PGD w/o LALR) leads to significantly poor TA and RA. As the 18 times larger batch size is used, DAT-PGD w/o LALR yields more than 25% drop in TA and 10% drop in RA compared to the best AT case. We find that the performance of DAT-PGD w/o LALR rapidly degrades as the number of computing nodes increases. The similar conclusion holds for DAT-FGSM w/o LALR vs. Fast AT. Furthermore, we

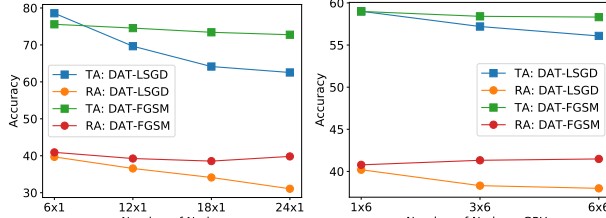

Figure 1: TA/RA comparison between DAT-FGSM and DAT-LSGD vs. node-GPU configurations. Left: (CIFAR-10, ResNet-18). Right: (ImageNet, ResNet-50).

observe that DAT-PGD outperforms DAT-LSGD (Xie et al., 2019) with 16.13% and 4.32% improvement in TA and RA, respectively. In Figure 1, we further compare our proposed DAT with the DAT-LSGD baseline in terms of TA/RA versus the number of computing nodes. Clearly, our approach scales more gracefully than the baseline (without losing much performance as the batch sizes increases along the number of computing nodes).

Moreover, we consistently observe that DAT-PGD (or DAT-FGSM) is able to achieve competitive performance to AT (or Fast AT) and enables a graceful training speedup, e.g., by 3 times using 6 machines for ImageNet. In practice, DAT is not able to achieve linear speed-up mainly because of the communication cost. For example, when comparing the computation time of DAT-PGD (batch size $6 \times 512$) with that of AT (batch size 512) under ImageNet, the computation speed-up (by excluding the communication cost) is given by $(6022)/(1960 - 898) = 5.67$, consistent with the ideal computation gain using $6\times$ larger batch size in DAT-PGD. Furthermore, we observe that when the largest batch size ($24 \times 2048$) is used, DAT-FGSM takes only (500 seconds) to obtain satisfactory robustness.

When comparing DAT-FGSM with DAT-PGD, we consistently observe that the former is capable of offering satisfactory (and even better) RA, but inevitably introduces a TA loss. This phenomenon also holds for Fast AT versus AT, e.g., 0.4% RA improvement vs. 3.71% TA degradation for ImageNet. We also note that the per-epoch communication time decreases when the more GPU machines (24) are used, since a larger batch size allows a smaller number of iterations per epoch, leading to less frequent communications among machines. In Appendix 4.3, we present additional results on CIFAR-10 using ResNet-50.

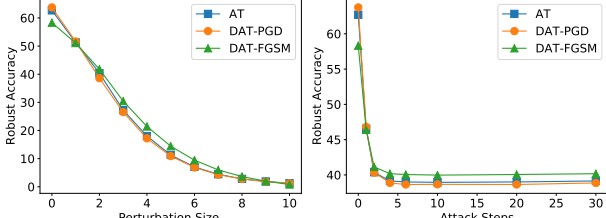

Figure 2: RA against PGD attacks for model trained by DAT-PGD, DAT-FGSM, and AT following (ImageNet, ResNet-50) in Table 1. (Left) RA versus different perturbation sizes (over the divisor 255). (Right) RA versus different steps.

**Robustness against different PGD attacks** In Figure 2, we evaluate the adversarial robustness of ResNet-50 at ImageNet learned by DAT-PGD and DAT-FGSM against PGD attacks of different steps and perturbation sizes (i.e., values of $\epsilon$). We observe that DAT matches robust accuracies of standard AT even against PGD attacks at different values of $\epsilon$ and steps. We also note that although DAT-FGSM has the worst TA ($\epsilon = 0$), it yields slightly better robustness as attack steps increase. The similar results can be found in Appendix 4.4 for (CIFAR-10, ResNet-18) against PGD (and C&W) attacks.

**DAT under unlabeled data** In Table 2, we report TA and RA of DAT in the semi-supervised setting (Carmon et al., 2019) with the use of 500K unlabeled images mined from Tiny Images (Carmon et al., 2019). Compared to the DAT supervised learning results at (CIFAR-10,

Table 2: DAT in semi-supervised learning with unlabeled data, where the computing resource configuration and batch size are set the same as the 8th and 10th rows of (CIFAR-10, ResNet-18) in Table 1. The relative improvement over RA or TA obtained in supervised learning (CIFAR-10 only) is marked by red color.

| Method | CIFAR-10 + 500K unlabeled Tiny Images, ResNet-18 | | | |
|---|---|---|---|---|
| | TA (%) | RA (%) | Comm. | Tr. time |
| DAT-PGD | 87.00 (↑ 7.62) | 47.34 (↑ 8.40) | 86 | 451 |
| DAT-FGSM | 88.00 (↑ 12.42) | 45.84 (↑ 4.92) | 86 | 124 |

ResNet-18) in Table 1, we observe that although the communication and computation costs increase due to the use of additional unlabeled images, both TA and RA are significantly improved. In particular, the performance of DAT-FGSM matches that of DAT-PGD. This suggests that unlabeled data might provide a solution to compensate the TA loss induced by FGSM-based robust training algorithms.

**DAT from pre-training to fine-tuning** In Figure 3, we investigate if a DAT pre-trained model (ResNet-50) over a source dataset (ImageNet) can offer a fast fine-tuning to a down-stream target dataset (CIFAR-100). Here we up-sample a CIFAR image to the same dimension of an ImageNet image before feeding it into the pre-trained model (Shafahi et al., 2020). Compared with the direct application of DAT to the target dataset (without pre-training), the pre-training enables a fast adaption to the down-stream CIFAR task in both TA and RA within just 3 epochs. Thus, the scalability of DAT to large datasets and multiple nodes offers a great potential in the pre-training + fine-tuning paradigm. The similar results can be found in Appendix 4.5 on CIFAR-10.

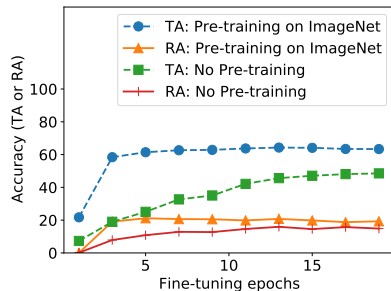

Figure 3: Fine-tuning ResNet-50 (pre-trained on ImageNet) under CIFAR-100. Here DAT-PGD is used for both pre-training and fine-tuning at 6 nodes with batch size $6 \times 128$.

**Quantization effect** In Appendix 4.6, we also study how the performance of DAT is affected by gradient quantization. We find that when the number of bits is reduced from $32$ to $8$, the resulting TA and RA becomes worse than the best 32-bit case. For example, in the worst case (8-bit 2-sided quantization) of CIFAR-10, TA drops $1.52\%$ and $6.32\%$ for DAT-PGD and DAT-FGSM, respectively. And RA drops $4.74\%$ and $5.58\%$, respectively. Note that our main communication configuration is given by Ring-AllReduce that calls for 1-sided (rather than 2-sided) quantization. We also observe that DAT-FGSM is more sensitive to effect of gradient quantization than DAT-PGD. Even in the centralized setting, the use of 8-bit quantization can lead to a non-trivial drop in TA (see Table A5). However, the use of quantization reduces the amount of data transmission per iteration. We also show that if a high performance computing cluster of nodes (with NVLink high-speed GPU interconnect (Foley & Danskin, 2017)) is used, the communication cost can be further reduced.

**LALR vs. centralized and distributed solutions** We examine the effect of LALR on both centralized and distributed robust training methods given a batch size that is affordable to a single machine. We consider a variant of AT by incorporating LALR, termed as AT w/ LALR. As presented in Table 3, when the batch size is not large, both centralized and distributed methods lead to very similar performance although the former is slightly better as it is free of machine synchronization and communication. And the performance is not sensitive to LALR. By contrast, if the batch size is large (inapplicable to centralized cases as Table 1), then DAT + LALR outperforms DAT (namely, LALR matters).

Table 3: Effect of LALR on centralized and distributed training under CIAR-10 with same batch size (2048).

| Method | # machines × batch size per machine | TA (%) | RA (%) |
|---|---|---|---|
| DAT-PGD (w/ LALR) | $6 \times 341$ | 83.48 | 39.76 |
| AT w/ LALR | $1 \times 2048$ | 83.56 | 39.84 |
| DAT w/o LALR | $6 \times 341$ | 82.42 | 38.41 |
| AT (w/o LALR) | $1 \times 2048$ | 82.94 | 38.54 |

## 6 CONCLUSIONS

We proposed distributed adversarial training (DAT) to scale up the training of adversarially robust DNNs over multiple machines. We showed that DAT is general in that it enables large-batch min-max optimization and supports gradient compression and different learning regimes. We proved that under mild conditions, DAT is guaranteed to converge to a first-order stationary point with a sub-linear rate. Empirically, we provided comprehensive experiment results to demonstrate the effectiveness and the usefulness of DAT in training robust DNNs with large datasets and multiple machines. In the future, it will be worthwhile to examine the speedup achieved by DAT in the extreme training cases, e.g., using a significantly large number of PGD attack steps and/or computing nodes during training.

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

APPENDIX

# 1 DAT ALGORITHM FRAMEWORK

---

**Algorithm A1** Distributed adversarial training (DAT) for solving problem (2)

---

1: **Input:** Initial $\boldsymbol{\theta}_1$, dataset $\mathcal{D}^{(i)}$ for each of $M$ workers, and $T$ iterations
2: **for** Iteration $t = 1, 2, \ldots, T$
3:      **for** Worker $i = 1, 2, \ldots, M$                                              ▷ Worker
4:          Draw a finite-size data batch $\mathcal{B}_t^i \subseteq \mathcal{D}^{(i)}$
5:          For each data sample $\mathbf{x} \in \mathcal{B}_t^i$, call for an *inner maximization oracle*:

$$\boldsymbol{\delta}_t^{(i)}(\mathbf{x}) := \arg\max_{\|\boldsymbol{\delta}\|_\infty \leq \epsilon} \phi(\boldsymbol{\theta}_t, \boldsymbol{\delta}; \mathbf{x}), \tag{A1}$$

where we omit the label or possible pseudo-label $y$ of $\mathbf{x}$ for brevity
6:          Computing local gradient of $f_i$ in (2) with respect to $\boldsymbol{\theta}$ given perturbed samples:

$$\mathbf{g}_t^{(i)} = \lambda \mathbb{E}_{\mathbf{x} \in \mathcal{B}_t^{(i)}} [\nabla_{\boldsymbol{\theta}} \ell(\boldsymbol{\theta}_t; \mathbf{x})] + \mathbb{E}_{\mathbf{x} \in \mathcal{B}_t^{(i)}} [\nabla_{\boldsymbol{\theta}} \phi(\boldsymbol{\theta}_t; \mathbf{x} + \boldsymbol{\delta}_t^{(i)}(\mathbf{x}))] \tag{A2}$$

7:          (*Optional*) Call for *gradient quantizer* $Q(\cdot)$ and transmit $Q(\mathbf{g}_t^{(i)})$ to server
8:      **end for**
9:      Gradient aggregation at server:                                              ▷ Server

$$\hat{\mathbf{g}}_t = \tfrac{1}{M} \sum_{i=1}^M Q(\mathbf{g}_t^{(i)}) \tag{A3}$$

10:      (*Optional*) Call for *gradient quantizer* $\hat{\mathbf{g}}_t \leftarrow Q(\hat{\mathbf{g}}_t)$, and transmit $\hat{\mathbf{g}}_t$ to workers:
11:      **for** Worker $i = 1, 2, \ldots, M$                                              ▷ Worker
12:          Call for an *outer minimization oracle* $\mathcal{A}(\cdot)$ to update $\boldsymbol{\theta}$:

$$\boldsymbol{\theta}_{t+1} = \mathcal{A}(\boldsymbol{\theta}_t \hat{\mathbf{g}}_t, \eta_t), \qquad \eta_t \text{ is learning rate} \tag{A4}$$

13:      **end for**
14: **end for**

---

**Additional details on gradient quantization** Let $b$ denote the number of bits ($b \leq 32$), and thus there exists $s = 2^b$ quantization levels. We specify the gradient quantization operation $Q(\cdot)$ in Algorithm A1 as the *randomized quantizer* (Alistarh et al., 2017; Yu et al., 2019). Formally, the quantization operation at the $i$th coordinate of a vector $\mathbf{g}$ is given by (Alistarh et al., 2017)

$$Q(g_i) = \|\mathbf{g}\|_2 \cdot \mathrm{sign}(g_i) \cdot \xi_i(g_i, s), \quad \forall i \in \{1, 2, \ldots, d\}. \tag{A5}$$

In (A5), $\xi_i(g_i, s)$ is a random number drawn as follows. Given $|g_i|/\|\mathbf{g}\|_2 \in [l/s, (l+1)/s]$ for some $l \in \mathbb{N}^+$ and $0 \leq l < s$, we then have

$$\xi_i(g_i, s) = \begin{cases} l/s & \text{with probability } 1 - (s|g_i|/\|\mathbf{g}\|_2 - l) \\ (l+1)/s & \text{with probability } (s|g_i|/\|\mathbf{g}\|_2 - l), \end{cases} \tag{A6}$$

where $|a|$ denotes the absolute value of a scalar $a$, and $\|\mathbf{a}\|_2$ denotes the $\ell_2$ norm of a vector $\mathbf{a}$. The rationale behind using (A5) is that $Q(g_i)$ is an *unbiased* estimate of $g_i$, namely, $\mathbb{E}_{\xi_i(g_i,s)}[Q(g_i)] = g_i$, with bounded variance. Moreover, we at most need $(32 + d + bd)$ bits to transmit the quantized $Q(\mathbf{g})$, where 32 bits for $\|\mathbf{g}\|_2$, 1 bit for sign of $g_i$ and $b$ bits for $\xi_i(g_i, s)$, whereas it needs $32d$ bits for a single-precision $\mathbf{g}$. Clearly, a small $b$ saves the communication cost. We note that if every worker performs as a server in DAT, then the quantization operation at Step 10 of Algorithm A1 is no longer needed. In this case, the communication network becomes fully connected. With synchronized communication, this is favored for training DNNs under the All-reduce operation.

## 2 THEORETICAL RESULTS

In this section, we will quantify the convergence behaviour of the proposed DAT algorithm. First, we define the following notations:

$$\Phi_i(\boldsymbol{\theta}, \mathbf{x}) = \max_{\|\boldsymbol{\delta}^{(i)}\|_\infty \leq \epsilon} \phi(\boldsymbol{\theta}, \boldsymbol{\delta}^{(i)}; \mathbf{x}), \quad \text{and} \quad \Phi_i(\boldsymbol{\theta}) = \mathbb{E}_{\mathbf{x} \in \mathcal{D}^{(i)}} \Phi_i(\boldsymbol{\theta}; \mathbf{x}). \tag{A7}$$

We also define

$$l_i(\boldsymbol{\theta}) = \mathbb{E}_{\mathbf{x} \in \mathcal{D}^{(i)}} l(\boldsymbol{\theta}; \mathbf{x}), \tag{A8}$$

where the label $y$ of $\mathbf{x}$ is omitted for labeled data. Then, the objective function of problem (2) can be expressed in the compact way

$$\Psi(\boldsymbol{\theta}) = \frac{1}{M} \sum_{i=1}^{M} \lambda l_i(\boldsymbol{\theta}) + \Phi_i(\boldsymbol{\theta}) \tag{A9}$$

and the optimization problem is then given by $\min_{\boldsymbol{\theta}} \Psi(\boldsymbol{\theta})$.

Therefore, it is clear that if a point $\boldsymbol{\theta}^\star$ satisfies

$$\|\nabla_{\boldsymbol{\theta}} \Psi(\boldsymbol{\theta}^\star)\| \leq \xi, \tag{A10}$$

then we say $\boldsymbol{\theta}^\star$ is a $\xi$ approximate first-order stationary point (FOSP) of problem (2).

Prior to delving into the convergence analysis of DAT, we make the following assumptions.

### 2.1 ASSUMPTIONS

A1. Assume objective function has layer-wise Lipschitz continuous gradients with constant $L_i$ for each layer

$$\|\nabla_i \Psi(\boldsymbol{\theta}_{\cdot, i}) - \nabla_i \Psi(\boldsymbol{\theta}'_{\cdot, i})\| \leq L_i \|\boldsymbol{\theta}_{\cdot, i} - \boldsymbol{\theta}'_{\cdot, i}\|, \forall i \in [h]. \tag{A11}$$

where $\nabla_i \Psi(\boldsymbol{\theta}_{\cdot, i})$ denotes the gradient w.r.t. the variables at the $i$th layer. Also, we assume that $\Psi(\boldsymbol{\theta})$ is lower bounded, i.e., $\Psi^\star := \min_{\boldsymbol{\theta}} \Psi(\boldsymbol{\theta}) > -\infty$ and bounded gradient estimate, i.e., $\|\nabla \hat{\mathbf{g}}_t^{(i)}\| \leq G$.

A2. Assume that $\phi(\boldsymbol{\theta}, \boldsymbol{\delta}; \mathbf{x})$ is strongly concave with respect to $\boldsymbol{\delta}$ with parameter $\mu$ and has the following gradient Lipschitz continuity with constant $L_\phi$:

$$\|\nabla_{\boldsymbol{\theta}} \phi(\boldsymbol{\theta}, \boldsymbol{\delta}; \mathbf{x}) - \nabla_{\boldsymbol{\theta}} \phi(\boldsymbol{\theta}, \boldsymbol{\delta}'; \mathbf{x})\| \leq L_\phi \|\boldsymbol{\delta} - \boldsymbol{\delta}'\|. \tag{A12}$$

A3. Assume that the gradient estimate is unbiased and has bounded variance, i.e.,

$$\mathbb{E}_{\mathbf{x} \in \mathcal{B}^{(i)}}[\nabla_{\boldsymbol{\theta}} l(\boldsymbol{\theta}; \mathbf{x})] = \nabla_{\boldsymbol{\theta}} l(\boldsymbol{\theta}), \forall i, \tag{A13}$$

$$\mathbb{E}_{\mathbf{x} \in \mathcal{B}^{(i)}}[\nabla_{\boldsymbol{\theta}} \Phi(\boldsymbol{\theta}; \mathbf{x})] = \nabla_{\boldsymbol{\theta}} \Phi(\boldsymbol{\theta}), \forall i, \tag{A14}$$

where recall that $\mathcal{B}^{(i)}$ denotes a data batch used at worker $i$, $\nabla_{\boldsymbol{\theta}} l(\boldsymbol{\theta}) := \frac{1}{M} \sum_{i=1}^{M} \nabla_{\boldsymbol{\theta}} l_i(\boldsymbol{\theta})$ and $\nabla_{\boldsymbol{\theta}} \Phi(\boldsymbol{\theta}) := \frac{1}{M} \sum_{i=1}^{M} \nabla_{\boldsymbol{\theta}} \Phi_i(\boldsymbol{\theta})$; and

$$\mathbb{E}_{\mathbf{x} \in \mathcal{B}^{(i)}} \|\nabla_{\boldsymbol{\theta}} l(\boldsymbol{\theta}; \mathbf{x}) - \nabla_{\boldsymbol{\theta}} l(\boldsymbol{\theta})\|^2 \leq \sigma^2, \forall i \tag{A15}$$

$$\mathbb{E}_{\mathbf{x} \in \mathcal{B}^{(i)}} \|\nabla_{\boldsymbol{\theta}} \Phi(\boldsymbol{\theta}; \mathbf{x}) - \nabla_{\boldsymbol{\theta}} \Phi(\boldsymbol{\theta})\|^2 \leq \sigma^2, \forall i. \tag{A16}$$

Further, we define a component-wise bounded variance of the gradient estimate

$$\mathbb{E}_{\mathbf{x} \in \mathcal{B}^{(i)}} \|[\nabla_{\boldsymbol{\theta}} l(\boldsymbol{\theta}; \mathbf{x})]_{jk} - [\nabla_{\boldsymbol{\theta}} l(\boldsymbol{\theta})]_{jk}\|^2 \leq \sigma_{jk}^2, \forall i, \tag{A17}$$

$$\mathbb{E}_{\mathbf{x} \in \mathcal{B}^{(i)}} \|[\nabla_{\boldsymbol{\theta}} \Phi(\boldsymbol{\theta}; \mathbf{x})]_{jk} - [\nabla_{\boldsymbol{\theta}} \Phi(\boldsymbol{\theta})]_{jk}\|^2 \leq \sigma_{jk}'^2, \forall i, \tag{A18}$$

where $j$ denotes the index of the layer, and $k$ denotes the index of entry at each layer. Under A3, we have $\sum_{j=1}^{h} \sum_{k=1}^{d_j} \max\{\sigma_{jk}^2, \sigma_{jk}'^2\} \leq \sigma^2$

A4. Assume that the component wise compression error has bounded variance

$$\mathbb{E}[(Q([\mathbf{g}^{(i)}(\boldsymbol{\theta})]_{jk}) - [\mathbf{g}^{(i)}(\boldsymbol{\theta})]_{jk})^2] \leq \delta_{jk}^2, \forall i. \tag{A19}$$

The assumption A4 is satisfied as the randomized quantization is used (Alistarh et al., 2017, Lemma 3.1).

## 2.2 ORACLE OF MAXIMIZATION

In practice, $\Phi_i(\boldsymbol{\theta}; \mathbf{x}), \forall i$ may not be obtained, since the inner loop needs to iterate by the infinite number of iterations to achieve the exact maximum point. Therefore, we allow some numerical error term resulted in the maximization step at (A1). This consideration makes the convergence analysis more realistic.

First, we have the following criterion to measure the closeness of the approximate maximizer to the optimal one.

**Definition 1.** *Under A2, if point $\boldsymbol{\delta}(\mathbf{x})$ satisfies*

$$\max_{\boldsymbol{\delta} \leq \|\epsilon\|} \langle \boldsymbol{\delta} - \boldsymbol{\delta}^*(\mathbf{x}), \nabla_{\boldsymbol{\delta}} \phi(\boldsymbol{\theta}, \boldsymbol{\delta}^*(\mathbf{x}); \mathbf{x}) \rangle \leq \varepsilon \tag{A20}$$

*then, it is a $\varepsilon$ approximate solution to $\boldsymbol{\delta}^*(\mathbf{x})$, where*

$$\boldsymbol{\delta}^*(\mathbf{x}) := \arg\max_{\boldsymbol{\delta}} \phi(\boldsymbol{\theta}, \boldsymbol{\delta}; \mathbf{x}). \tag{A21}$$

*and $\mathbf{x}$ denotes the sampled data.*

Condition (A20) is standard for defining approximate solutions of an optimization problem over a compact feasible set and has been widely studied in (Wang et al., 2019b; Lu et al., 2020).

In the following, we can show that when the inner maximization problem is solved accurately enough, the gradients of function $\phi(\boldsymbol{\theta}, \boldsymbol{\delta}(\mathbf{x}); \mathbf{x})$ at $\boldsymbol{\delta}(\mathbf{x})$ and $\boldsymbol{\delta}^*(\mathbf{x})$ are also close. A similar claim of this fact has been shown in (Wang et al., 2019b, Lemma 2). For completeness of the analysis, we provide the specific statement for our problem here and give the detailed proof as well.

**Lemma 1.** *Let $\boldsymbol{\delta}_t^{(k)}$ be the $(\mu\varepsilon)/L_\phi^2$ approximate solution of the inner maximization problem for worker $k$, i.e., $\max_{\boldsymbol{\delta}^{(k)}} \phi(\boldsymbol{\theta}, \boldsymbol{\delta}^{(k)}; \mathbf{x}_t)$, where $\mathbf{x}_t$ denotes the sampled data at the $t$th iteration of DAT. Under A2, we have*

$$\left\| \nabla_{\boldsymbol{\theta}} \phi\left( \boldsymbol{\theta}_t, \boldsymbol{\delta}_t^{(k)}(\mathbf{x}_t); \mathbf{x}_t \right) - \nabla_{\boldsymbol{\theta}} \phi\left( \boldsymbol{\theta}_t, (\boldsymbol{\delta}^*)_t^{(k)}(\mathbf{x}_t); \mathbf{x}_t \right) \right\|^2 \leq \varepsilon. \tag{A22}$$

Throughout the convergence analysis, we assume that $\boldsymbol{\delta}_t^{(k)}(\mathbf{x}_t), \forall k, t$ are all the $(\mu\varepsilon)/L_\phi^2$ approximate solutions of the inner maximization problem. Let us define

$$\left\| [\nabla\phi(\boldsymbol{\theta}_t, \boldsymbol{\delta}_t^{(k)}(\mathbf{x}_t); \mathbf{x}_t)]_{ij} - [\nabla\phi(\boldsymbol{\theta}_t, (\boldsymbol{\delta}^*)_t^{(k)}(\mathbf{x}_t); \mathbf{x}_t]_{ij} \right\|^2 = \varepsilon_{ij}. \tag{A23}$$

From Lemma 1, we know that when $\boldsymbol{\delta}_t^{(k)}(\mathbf{x}_t)$ is a $(\mu\varepsilon)/L_\phi^2$ approximate solution, then

$$\sum_{i=1}^h \sum_{j=1}^{d_i} \varepsilon_{ij} = \sum_{i=1}^h \sum_{j=1}^{d_i} \left\| [\nabla\phi(\boldsymbol{\theta}_t, \boldsymbol{\delta}_t^{(k)}(\mathbf{x}_t); \mathbf{x}_t)]_{ij} - [\nabla\phi(\boldsymbol{\theta}_t, (\boldsymbol{\delta}^*)_t^{(k)}(\mathbf{x}_t); \mathbf{x}_t]_{ij} \right\|^2 \leq \varepsilon. \tag{A24}$$

## 2.3 FORMAL STATEMENTS OF CONVERGENCE RATE GUARANTEES

In what follows, we provide the formal statement of convergence rate of DAT. In our analysis, we focus on the 1-sided quantization, namely, Step 10 of Algorithm A1 is omitted, and specify the outer minimization oracle by LAMB (You et al., 2019), see Algorithm A2. The addition and multiplication operations in LAMB are component-wise.

**Theorem 2.** *Under A1-A4, suppose that $\{\boldsymbol{\theta}_t\}$ is generated by DAT for a total number of $T$ iterations, and let the problem dimension at each layer be $d_i = d/h$. Then the convergence rate of DAT is given by*

$$\frac{1}{T} \sum_{t=1}^T \mathbb{E} \|\nabla_{\boldsymbol{\theta}} \Psi(\boldsymbol{\theta}_t)\|^2 \leq \frac{\Delta_\Psi}{\eta_t c_l C T} + 2\left(\varepsilon + \frac{(1+\lambda)\sigma^2}{MB}\right) + 4\delta^2 + \frac{\kappa\sqrt{3}}{C}\|\boldsymbol{\chi}\|_1 + \frac{\eta_t c_u \kappa \|L\|_1}{2C}. \tag{A25}$$

*where $\Delta_\Psi := \mathbb{E}[\Psi(\boldsymbol{\theta}_1)] - \Psi^\star]$, $\eta_t$ is the learning rate, $\kappa = c_u/c_l$, $c_l$ and $c_u$ are constants used in LALR (5), $\boldsymbol{\chi}$ is an error term with the $(ih + j)$th entry being $\sqrt{\frac{(1+\lambda)\sigma_{ij}^2}{MB} + \varepsilon_{ij} + \delta_{ij}^2}$, $\varepsilon$ and*

$\varepsilon_{ij}$ were given in (A24), $L = [L_1, \ldots, L_h]^T$, $C = \frac{1}{4}\sqrt{\frac{h(1-\beta_2)}{G^2 d}}$, $0 < \beta_2 < 1$ is given in LAMB, $B = \min\{|\mathcal{B}^{(i)}|, \forall i\}$, and $G$ is given in A1.

*Remark 1.* When the batch size is large, i.e., $B \sim \sqrt{T}$, then the gradient estimate error will be $\mathcal{O}(\sigma^2/\sqrt{T})$. Further, it is worth noting that different from the convergence results of LAMB, there is a linear speedup of deceasing the gradient estimate error in DAT with respect to $M$, i.e., $\mathcal{O}(\sigma^2/(M\sqrt{T}))$, which is the advantage of using multiple computing nodes.

*Remark 2.* Note that A4 implies $\mathbb{E}[(Q([\mathbf{g}^{(k)}(\boldsymbol{\theta})]_{ij}) - [\mathbf{g}^{(k)}(\boldsymbol{\theta})]_{ij}\|^2] \leq \sum_{i=1}^{h}\sum_{j=1}^{d_i}\delta_{ij}^2 := \delta^2$. From (Alistarh et al., 2017, Lemma 3.1), we know that $\delta^2 \leq \min\{d/s^2, \sqrt{d}/s\}G^2$. Recall that $s = 2^b$, where $b$ is the number of quantization bits.

Therefore, with a proper choice of the parameters, we can have the following convergence result that has been shown in Theorem 1.

**Corollary 1.** *Under the same conditions of Theorem 2, if we choose*

$$\eta_t \sim \mathcal{O}(1/\sqrt{T}), \quad \varepsilon \sim \mathcal{O}(\xi^2), \tag{A26}$$

*we then have*

$$\frac{1}{T}\sum_{t=1}^{T}\mathbb{E}\|\nabla_{\boldsymbol{\theta}}\Psi(\boldsymbol{\theta}_t)\|^2 \leq \frac{\Delta_{\Psi}}{c_l C\sqrt{T}} + \frac{(1+\lambda)\sigma^2}{MB} + \frac{c_u\kappa\|L\|_1}{2C\sqrt{T}} + \mathcal{O}\left(\xi, \frac{\sigma}{\sqrt{MT}}, \min\left\{\frac{d}{4^b}, \frac{\sqrt{d}}{2^b}\right\}\right). \tag{A27}$$

In summary, when the batch size is large enough, DAT converges to a first-order stationary point of problem (2) and there is a linear speed-up in terms of $M$ with respect to $\sigma^2$. Next, we provide the details of the proof.

# 3 DETAILED PROOFS OF CONVERGENCE ANALYSIS

## 3.1 PRELIMINARIES

In the proof, we use the following inequality and notations.

1. Young's inequality with parameter $\epsilon$ is

$$\langle \mathbf{x}, \mathbf{y} \rangle \leq \frac{1}{2\epsilon}\|\mathbf{x}\|^2 + \frac{\epsilon}{2}\|\mathbf{y}\|^2, \tag{A28}$$

where $\mathbf{x}, \mathbf{y}$ are two vectors.

2. Define the historical trajectory of the iterates as $\mathcal{F}_t = \{\boldsymbol{\theta}_{t-1}, \ldots, \boldsymbol{\theta}_1\}$.

3. We denote vector $[\mathbf{x}]_i$ as the parameters at the $i$th layer of the neural net and $[\mathbf{x}]_{ij}$ represents the $j$th entry of the parameter at the $i$th layer.

4. We define

$$\mathbf{g}_t := \frac{1}{M}\sum_{i=1}^{M}\mathbb{E}_{\mathbf{x}_t \in \mathcal{B}^{(i)}}\left(\lambda\nabla l(\boldsymbol{\theta}_t; \mathbf{x}_t) + \nabla_{\boldsymbol{\theta}}\phi(\boldsymbol{\theta}_t, \boldsymbol{\delta}_t^{(i)}(\mathbf{x}_t); \mathbf{x}_t)\right) = \frac{1}{M}\sum_{i=1}^{M}\mathbf{g}_t^{(i)}. \tag{A29}$$

## 3.2 Details of LAMB algorithm

---

**Algorithm A2** LAMB (You et al., 2019)

---

**Input:** learning rate $\eta_t$, $0 < \beta_1, \beta_2 < 1$, scaling function $\tau(\cdot)$, $\zeta > 0$
**for** $t = 1, \ldots$ **do**
    $\mathbf{m}_t = \beta_1 \mathbf{m}_{t-1} + (1 - \beta_1)\hat{\mathbf{g}}_t$, where $\hat{\mathbf{g}}_t$ is given by (A3)
    $\mathbf{v}_t = \beta_2 \mathbf{v}_{t-1} + (1 - \beta_2)\hat{\mathbf{g}}_t^2$
    $\mathbf{m}_t = \mathbf{m}_t/(1 - \beta_1^t)$
    $\mathbf{v}_t = \mathbf{v}_t/(1 - \beta_2^t)$
    Compute ratio $\mathbf{u}_t = \frac{\mathbf{m}_t}{\sqrt{\mathbf{v}_t} + \zeta}$
**end for**
Update

$$\boldsymbol{\theta}_{t+1,i} = \boldsymbol{\theta}_{t,i} - \frac{\eta_t \tau(\|\boldsymbol{\theta}_{t,i}\|)}{\|\mathbf{u}_{t,i}\|} \mathbf{u}_{t,i}. \tag{A30}$$

---

## 3.3 Proof of Lemma 1

*Proof.* From A2, we have

$$\left\| \nabla\phi\left(\boldsymbol{\theta}_t, \boldsymbol{\delta}_t^{(i)}(\mathbf{x}_t); \mathbf{x}_t\right) - \nabla\phi\left(\boldsymbol{\theta}_t, (\boldsymbol{\delta}^*)_t^{(i)}(\mathbf{x}_t); \mathbf{x}_t\right) \right\| \leq L_\phi \|\boldsymbol{\delta}_t^{(i)}(\mathbf{x}_t) - (\boldsymbol{\delta}^*)_t^{(i)}(\mathbf{x}_t)\|. \tag{A31}$$

Also, we know that function $\phi(\boldsymbol{\theta}, \boldsymbol{\delta}, \mathbf{x})$ is strongly concave with respect to $\boldsymbol{\delta}$, so we have

$$\mu\|\boldsymbol{\delta}_t^{(i)}(\mathbf{x}_t) - (\boldsymbol{\delta}^*)_t^{(i)}(\mathbf{x}_t)\|$$
$$\leq \left\langle \nabla_{\boldsymbol{\delta}}\phi(\boldsymbol{\theta}_t, (\boldsymbol{\delta}^*)_t^{(i)}(\mathbf{x}_t); \mathbf{x}_t) - \nabla_{\boldsymbol{\delta}}\phi(\boldsymbol{\theta}_t, \boldsymbol{\delta}_t^{(i)}(\mathbf{x}_t); \mathbf{x}_t), \boldsymbol{\delta}_t^{(i)}(\mathbf{x}_t) - (\boldsymbol{\delta}^*)_t^{(i)}(\mathbf{x}_t) \right\rangle. \tag{A32}$$

Next, we have two conditions about the qualities of solutions $\boldsymbol{\delta}_t^{(i)}(\mathbf{x}_t)$ and $(\boldsymbol{\delta}^*)_t^{(i)}(\mathbf{x}_t)$. First, we know that $\boldsymbol{\delta}_t^{(i)}(\mathbf{x}_t)$ is a-$\varepsilon$ approximate solution to $(\boldsymbol{\delta}^*)_t^{(i)}(\mathbf{x}_t)$, so we have

$$\left\langle (\boldsymbol{\delta}^*)_t^{(i)}(\mathbf{x}_t) - \boldsymbol{\delta}_t^{(i)}(\mathbf{x}_t), \nabla_{\boldsymbol{\delta}}\phi(\boldsymbol{\theta}_t, \boldsymbol{\delta}_t^{(i)}(\mathbf{x}_t); \mathbf{x}_t) \right\rangle \leq \varepsilon. \tag{A33}$$

Second, since $(\boldsymbol{\delta}^*)_t^{(i)}(\mathbf{x}_t)$ is the optimal solution, it satisfies

$$\left\langle (\boldsymbol{\delta}_t^{(i)}(\mathbf{x}_t) - (\boldsymbol{\delta}^*)_t^{(i)}(\mathbf{x}_t), \nabla_{\boldsymbol{\delta}}\phi(\boldsymbol{\theta}_t, (\boldsymbol{\delta}^*)_t^{(i)}(\mathbf{x}_t); \mathbf{x}_t) \right\rangle \leq 0. \tag{A34}$$

Adding them together, we can obtain

$$\left\langle \boldsymbol{\delta}_t^{(i)}(\mathbf{x}_t) - (\boldsymbol{\delta}^*)_t^{(i)}(\mathbf{x}_t), \nabla_{\boldsymbol{\delta}}\phi(\boldsymbol{\theta}_t, (\boldsymbol{\delta}^*)_t^{(i)}(\mathbf{x}_t); \mathbf{x}_t) - \nabla_{\boldsymbol{\delta}}\phi(\boldsymbol{\theta}_t, \boldsymbol{\delta}_t^{(i)}(\mathbf{x}_t); \mathbf{x}_t) \right\rangle \leq \varepsilon. \tag{A35}$$

Substituting (A35) into (A32), we can get

$$\mu\|\boldsymbol{\delta}_t^{(i)}(\mathbf{x}_t) - (\boldsymbol{\delta}^*)_t^{(i)}(\mathbf{x}_t)\|^2 \leq \varepsilon. \tag{A36}$$

Combining (A31), we have

$$\left\| \nabla\phi(\boldsymbol{\theta}_t, \boldsymbol{\delta}_t^{(i)}(\mathbf{x}_t); \mathbf{x}_t) - \nabla\phi(\boldsymbol{\theta}_t, (\boldsymbol{\delta}^*)_t^{(i)}(\mathbf{x}_t); \mathbf{x}_t) \right\|^2 \leq L_\phi^2 \frac{\varepsilon}{\mu}. \tag{A37}$$

$\square$

## 3.4 Descent of quantized LAMB

First, we provide the following lemma as a stepping stone for the subsequent analysis.

**Lemma 2.** *Under A1–A3, suppose that sequence $\{\boldsymbol{\theta}_t\}$ is generated by DAT. Then, we have*

$$\mathbb{E}[-\langle \nabla\Psi(\boldsymbol{\theta}_t), \hat{\mathbf{g}}_t \rangle] \leq -\frac{\mathbb{E}\|\nabla\Psi(\boldsymbol{\theta}_t)\|^2}{2} + \varepsilon + \frac{(1+\lambda)\sigma^2}{MB}. \tag{A38}$$

*Proof.* From (A21), (A7) and A2, we know that

$$\nabla_{\boldsymbol{\theta}}\Phi_i(\boldsymbol{\theta},\mathbf{x}) = \nabla_{\boldsymbol{\theta}}\phi(\boldsymbol{\theta},(\boldsymbol{\delta}^*)^{(i)}(\mathbf{x});\mathbf{x}), \tag{A39}$$

so we can get

$$\nabla_{\boldsymbol{\theta}}\Psi(\boldsymbol{\theta}) = \frac{1}{M}\sum_{i=1}^{M}\lambda\nabla_{\boldsymbol{\theta}}l_i(\boldsymbol{\theta}) + \nabla_{\boldsymbol{\theta}}\Phi_i(\boldsymbol{\theta}) \tag{A40}$$

$$= \lambda\nabla_{\boldsymbol{\theta}}l(\boldsymbol{\theta}) + \frac{1}{M}\sum_{i=1}^{M}\mathbb{E}_{\mathbf{x}\in\mathcal{D}^{(i)}}\nabla_{\boldsymbol{\theta}}\phi(\boldsymbol{\theta},(\boldsymbol{\delta}^*)^{(i)}(\mathbf{x});\mathbf{x}) \tag{A41}$$

$$:= \bar{\mathbf{g}}(\boldsymbol{\theta}). \tag{A42}$$

Then, we have

$$\mathbb{E}\langle\nabla\Psi(\boldsymbol{\theta}_t),\mathbf{g}_t\rangle = \mathbb{E}\langle\nabla\Psi(\boldsymbol{\theta}_t),\bar{\mathbf{g}}_t\rangle + \mathbb{E}\langle\nabla\Psi(\boldsymbol{\theta}_t),\mathbf{g}_t - \bar{\mathbf{g}}_t\rangle \tag{A43}$$

$$= \mathbb{E}_{\mathcal{F}_t}\mathbb{E}_{\mathbf{x}_t|\mathcal{F}_t}\langle\nabla\Psi(\boldsymbol{\theta}_t),\bar{\mathbf{g}}_t\rangle + \mathbb{E}\langle\nabla\Psi(\boldsymbol{\theta}_t),\mathbf{g}_t - \bar{\mathbf{g}}_t\rangle \tag{A44}$$

$$\overset{(A42)}{=} \mathbb{E}\|\nabla\Psi(\boldsymbol{\theta}_t)\|^2 + \mathbb{E}\langle\nabla\Psi(\boldsymbol{\theta}_t),\mathbf{g}_t - \bar{\mathbf{g}}_t\rangle \tag{A45}$$

$$= \mathbb{E}\|\nabla\Psi(\boldsymbol{\theta}_t)\|^2 + \mathbb{E}\langle\nabla\Psi(\boldsymbol{\theta}_t),\mathbf{g}_t - \mathbf{g}_t^*\rangle + \mathbb{E}\langle\nabla\Psi(\boldsymbol{\theta}_t),\mathbf{g}_t^* - \bar{\mathbf{g}}_t\rangle \tag{A46}$$

where

$$\bar{\mathbf{g}}_t := \frac{1}{M}\sum_{i=1}^{M}\mathbb{E}_{\mathbf{x}_t\in\mathcal{D}^{(i)}}\left(\lambda\nabla l(\boldsymbol{\theta}_t,\mathbf{x}_t) + \nabla_{\boldsymbol{\theta}}\phi(\boldsymbol{\theta}_t,(\boldsymbol{\delta}^*)_t^{(i)}(\mathbf{x}_t);\mathbf{x}_t)\right) = \lambda\nabla l(\boldsymbol{\theta}_t) + \nabla\Phi(\boldsymbol{\theta}_t), \tag{A47}$$

and

$$\mathbf{g}_t^* := \frac{1}{M}\sum_{i=1}^{M}\mathbb{E}_{\mathbf{x}_t\in\mathcal{B}^{(i)}}\left(\lambda\nabla l(\boldsymbol{\theta}_t,\mathbf{x}_t) + \nabla_{\boldsymbol{\theta}}\phi(\boldsymbol{\theta}_t,(\boldsymbol{\delta}^*)_t^{(i)}(\mathbf{x}_t);\mathbf{x}_t)\right). \tag{A48}$$

Next, we can quantify the different between $\mathbf{g}_t$ and $\mathbf{g}_t^*$ by gradient Lipschitz continuity of function $\tau(\cdot)$ as the following

$$\mathbb{E}\|\mathbf{g}_t - \mathbf{g}_t^*\|^2 \overset{(a)}{\leq} \frac{1}{M}\sum_{i=1}^{M}\mathbb{E}_{\mathcal{F}_t}\mathbb{E}_{\mathbf{x}_t|\mathcal{F}_t}\left[\|\nabla_{\boldsymbol{\theta}}\phi(\boldsymbol{\theta}_t,(\boldsymbol{\delta}^*)^{(i)}(\mathbf{x}_t);\mathbf{x}_t) - \nabla_{\boldsymbol{\theta}}\phi(\boldsymbol{\theta}_t,\boldsymbol{\delta}^{(i)}(\mathbf{x}_t);\mathbf{x}_t)\|^2\right] \overset{(A24)}{\leq} \varepsilon \tag{A49}$$

where in $(a)$ we use Jensen's inequality.

And the difference between $\bar{\mathbf{g}}_t$ and $\mathbf{g}_t^*$ can be upper bounded by

$$\mathbb{E}\|\bar{\mathbf{g}}_t - \mathbf{g}_t^*\|^2 = \mathbb{E}_{\mathcal{F}_t}\left\|\frac{1}{M}\sum_{i=1}^{M}\mathbb{E}_{\mathbf{x}_t|\mathcal{F}_t}\nabla_{\boldsymbol{\theta}}\phi(\boldsymbol{\theta}_t,(\boldsymbol{\delta}^*)^{(i)}(\mathbf{x}_t);\mathbf{x}_t) - \nabla_{\boldsymbol{\theta}}\phi(\boldsymbol{\theta}_t)\right\|^2$$

$$+ \lambda\mathbb{E}_{\mathcal{F}_t}\left\|\frac{1}{M}\sum_{i=1}^{M}\mathbb{E}_{\mathbf{x}_t|\mathcal{F}_t}\nabla l(\boldsymbol{\theta}_t;\mathbf{x}_t) - \nabla l(\boldsymbol{\theta}_t)\right\|^2 \tag{A50}$$

$$\overset{A3}{=} \frac{(1+\lambda)\sigma^2}{MB}. \tag{A51}$$

Applying Young's inequality with parameter 2, we have

$$\mathbb{E}[-\langle\nabla\Psi(\boldsymbol{\theta}_t),\mathbf{g}_t\rangle] \leq -\mathbb{E}\|\nabla\Psi(\boldsymbol{\theta}_t)\|^2 + \frac{\mathbb{E}\|\nabla\Psi(\boldsymbol{\theta}_t)\|^2}{2} + \mathbb{E}\|\bar{\mathbf{g}}_t - \mathbf{g}_t^*\|^2 + \mathbb{E}\|\mathbf{g}_t^* - \mathbf{g}_t\|^2 \tag{A52}$$

$$\overset{(A49)}{\leq} -\frac{\mathbb{E}\|\nabla\Psi(\boldsymbol{\theta}_t)\|^2}{2} + \varepsilon + \frac{(1+\lambda)\sigma^2}{MB}. \tag{A53}$$

$$\square$$

### 3.5 PROOF OF THEOREM 2

*Proof.* We set $\beta_1 = 0$ in LAMB for simplicity. From gradient Lipschitz continuity, we have

$$\Psi(\boldsymbol{\theta}_{t+1}) \overset{A1}{\leq} \Psi(\boldsymbol{\theta}_t) + \sum_{i=1}^{h} \langle [\nabla_{\boldsymbol{\theta}} \Psi(\boldsymbol{\theta}_t)]_i, \boldsymbol{\theta}_{t+1,i} - \boldsymbol{\theta}_{t,i} \rangle + \sum_{i=1}^{h} \frac{L_i}{2} \|\boldsymbol{\theta}_{t+1,i} - \boldsymbol{\theta}_{t,i}\|^2 \tag{A54}$$

$$\overset{(a)}{\leq} \Psi(\boldsymbol{\theta}_t) \underbrace{-\eta_t \sum_{i=1}^{h} \sum_{j=1}^{d_i} \tau(\|\boldsymbol{\theta}_{t,i}\|) \left\langle [\nabla \Psi(\boldsymbol{\theta}_t)]_{ij}, \frac{[\mathbf{u}_t]_{ij}}{\|\mathbf{u}_{t,i}\|} \right\rangle}_{:=\mathcal{R}} + \sum_{i=1}^{h} \frac{\eta_t^2 c_u^2 L_i}{2}, \tag{A55}$$

where in $(a)$ we use (A30), and the upper bound of $\tau(\|\boldsymbol{\theta}_{t,i}\|)$.

Next, we split term $R$ as two parts by leveraging $\text{sign}([\nabla \Psi(\boldsymbol{\theta}_t)]_{ij})$ and $\text{sign}([\mathbf{u}_t]_{ij})$ as follows.

$$\mathcal{R} = -\eta_t \sum_{i=1}^{h} \sum_{j=1}^{d_i} \tau(\|\boldsymbol{\theta}_{t,i}\|) [\nabla \Psi(\boldsymbol{\theta}_t)]_{ij} \frac{[\mathbf{u}_t]_{ij}}{\|\mathbf{u}_{t,i}\|} \mathbb{1} \left( \text{sign}([\nabla \Psi(\boldsymbol{\theta}_t)]_{ij}) = \text{sign}([\mathbf{u}_t]_{ij}) \right)$$

$$- \eta_t \sum_{i=1}^{h} \sum_{j=1}^{d_i} \tau(\|\boldsymbol{\theta}_{t,i}\|) [\nabla \Psi(\boldsymbol{\theta}_t)]_{ij} \frac{[\mathbf{u}_t]_{ij}}{\|\mathbf{u}_{t,i}\|} \mathbb{1} \left( \text{sign}([\nabla \Psi(\boldsymbol{\theta}_t)]_{ij}) \neq \text{sign}([\mathbf{u}_t]_{ij}) \right) \tag{A56}$$

$$\overset{(a)}{\leq} -\eta_t c_l \sum_{i=1}^{h} \sum_{j=1}^{d_i} \sqrt{\frac{1-\beta_2}{G^2 d_i}} [\nabla \Psi(\boldsymbol{\theta}_t)]_{ij} [\hat{\mathbf{g}}_t]_{ij} \mathbb{1} \left( \text{sign}([\nabla [\Psi(\boldsymbol{\theta}_t)]_{ij}) = \text{sign}([\hat{\mathbf{g}}_t]_{ij}) \right)$$

$$- \eta_t \sum_{i=1}^{h} \sum_{j=1}^{d_i} \tau(\|\boldsymbol{\theta}_{t,i}\|) [\nabla \Psi(\boldsymbol{\theta}_t)]_{ij} \frac{[\mathbf{u}_t]_{ij}}{\|\mathbf{u}_{t,i}\|} \mathbb{1} \left( \text{sign}([\nabla \Psi(\boldsymbol{\theta}_t)]_{ij}) \neq \text{sign}([\mathbf{u}_t]_{ij}) \right) \tag{A57}$$

$$\overset{(b)}{\leq} -\eta_t c_l \sum_{i=1}^{h} \sum_{j=1}^{d_i} \sqrt{\frac{1-\beta_2}{G^2 d_i}} [\nabla \Psi(\boldsymbol{\theta}_t)]_{ij} [\hat{\mathbf{g}}_t]_{ij}$$

$$- \eta_t \sum_{i=1}^{h} \sum_{j=1}^{d_i} \tau(\|\boldsymbol{\theta}_{t,i}\|) [\nabla \Psi(\boldsymbol{\theta}_t)]_{ij} \frac{[\mathbf{u}_t]_{ij}}{\|\mathbf{u}_{t,i}\|} \mathbb{1} \left( \text{sign}([\nabla \Psi(\boldsymbol{\theta}_t)]_{ij}) \neq \text{sign}([\mathbf{u}_t]_{ij}) \right). \tag{A58}$$

where in $(a)$ we use the fact that $\|\mathbf{u}_{t,i}\| \leq \sqrt{\frac{d_i}{1-\beta_2}}$ and $\sqrt{\mathbf{v}_t} \leq G$, and in $(b)$ we add

$$-\eta_t c_l \sum_{i=1}^{h} \sum_{j=1}^{d_i} \sqrt{\frac{1-\beta_2}{G^2 d_i}} [\nabla \Psi(\boldsymbol{\theta}_t)]_{ij} [\hat{\mathbf{g}}_t]_{ij} \mathbb{1} \left( \text{sign}([\nabla \Psi(\boldsymbol{\theta}_t)]_{ij}) \neq \text{sign}([\hat{\mathbf{g}}_t]_{ij}) \right) \geq 0. \tag{A59}$$

Taking expectation on both sides of (A58), we have the following:

$$\mathbb{E}[\mathcal{R}] \leq \underbrace{-\eta_t c_l \sqrt{\frac{h(1-\beta_2)}{G^2 d}} \sum_{i=1}^{h} \sum_{j=1}^{d_i} \mathbb{E}[[\nabla \Psi(\boldsymbol{\theta}_t)]_{ij} [\hat{\mathbf{g}}_t]_{ij}}_{:=\mathcal{U}}$$

$$+ \underbrace{\eta_t c_u \sum_{i=1}^{h} \sum_{j=1}^{d_i} \mathbb{E}\left[ [\nabla \Psi(\boldsymbol{\theta}_t)]_{ij} \mathbb{1} \left( \text{sign}([\nabla \Psi(\boldsymbol{\theta}_t)]_{ij}) \neq \text{sign}([\mathbf{u}_t]_{ij}) \right) \right]}_{:=\mathcal{V}}. \tag{A60}$$

Next, we will get the upper bounds of $\mathcal{U}$ and $\mathcal{V}$ separably as follows. First, we write the inner product between $[\nabla\Psi(\boldsymbol{\theta})]_{ij}$ and $[\hat{\mathbf{g}}_t]_{ij}$ more compactly,

$$\mathcal{U} \leq -\eta_t c_l \sqrt{\frac{h(1-\beta_2)}{G^2 d}} \sum_{i=1}^{h} \mathbb{E}\left\langle [\nabla\Psi(\boldsymbol{\theta})]_i, [\hat{\mathbf{g}}_t]_i \right\rangle \tag{A61}$$

$$\leq -\eta_t c_l \sqrt{\frac{h(1-\beta_2)}{G^2 d}} \sum_{i=1}^{h} \mathbb{E}\left\langle [\nabla\Psi(\boldsymbol{\theta}_t)]_i, [\hat{\mathbf{g}}_t]_i - [\mathbf{g}_t]_i + [\mathbf{g}_t]_i \right\rangle \tag{A62}$$

$$\leq -\eta_t c_l \sqrt{\frac{h(1-\beta_2)}{G^2 d}} \left( \mathbb{E}\left\langle \nabla\Psi(\boldsymbol{\theta}), \mathbf{g}_t \right\rangle + \sum_{i=1}^{h} \mathbb{E}\left\langle [\nabla\Psi(\boldsymbol{\theta}_t)]_i, [\hat{\mathbf{g}}_t]_i - [\mathbf{g}_t]_i \right\rangle \right). \tag{A63}$$

Applying Lemma 2, we can get

$$\mathcal{U} \overset{(A38)}{\leq} -\eta_t c_l \sqrt{\frac{h(1-\beta_2)}{G^2 d}} \frac{1}{2} \mathbb{E}\|\nabla\Psi(\boldsymbol{\theta}_t)\|^2 + \eta_t c_l \sqrt{\frac{h(1-\beta_2)}{G^2 d}} \left( \varepsilon + \frac{(1+\lambda)\sigma^2}{MB} \right)$$
$$- \eta_t c_l \sqrt{\frac{h(1-\beta_2)}{G^2 d}} \sum_{i=1}^{h} \mathbb{E}\left\langle [\nabla\Psi(\boldsymbol{\theta}_t)]_i, [\hat{\mathbf{g}}_t]_i - [\mathbf{g}_t]_i \right\rangle \tag{A64}$$

$$\overset{(a)}{\leq} -\eta_t c_l \sqrt{\frac{h(1-\beta_2)}{G^2 d}} \frac{1}{2} \mathbb{E}\|\nabla\Psi(\boldsymbol{\theta}_t)\|^2 + \eta_t c_l \sqrt{\frac{h(1-\beta_2)}{G^2 d}} \left( \varepsilon + \frac{(1+\lambda)\sigma^2}{MB} \right)$$
$$+ \frac{\eta_t c_l}{4} \sqrt{\frac{h(1-\beta_2)}{G^2 d}} \mathbb{E}\|\nabla\Psi(\boldsymbol{\theta}_t)\|^2 + c_l \eta_t \sqrt{\frac{h(1-\beta_2)}{G^2 d}} \mathbb{E}\|\hat{\mathbf{g}}_t - \mathbf{g}_t\|^2 \tag{A65}$$

$$\overset{(b)}{\leq} -\frac{\eta_t c_l}{4} \sqrt{\frac{h(1-\beta_2)}{G^2 d}} \frac{1}{2} \mathbb{E}\|\nabla\Psi(\boldsymbol{\theta}_t)\|^2 + \eta_t c_l \sqrt{\frac{h(1-\beta_2)}{G^2 d}} \left( \varepsilon + \frac{(1+\lambda)\sigma^2}{MB} \right)$$
$$+ \eta_t c_l \sqrt{\frac{h(1-\beta_2)}{G^2 d}} \delta^2 \tag{A66}$$

where we use the in $(a)$ we use Young's inequality (with parameter 2), and in $(b)$ we have

$$\mathbb{E}\|\hat{\mathbf{g}}_t - \mathbf{g}_t\|^2 = \mathbb{E}\left\| \frac{1}{M} \sum_{i=1}^{M} Q(\mathbf{g}_t^{(i)}) - \mathbf{g}_t^{(i)} \right\|^2 \overset{A4}{\leq} \delta^2. \tag{A67}$$

Second, we give the upper of $\mathcal{V}$:

$$\mathcal{V} \leq \eta_t c_u \sum_{i=1}^{h} \sum_{j=1}^{d_i} [\nabla\Psi(\boldsymbol{\theta}_t)]_{ij} \underbrace{\mathbb{P}\left( \text{sign}([\nabla\Psi(\boldsymbol{\theta}_t)]_{ij}) \neq \text{sign}([\hat{\mathbf{g}}_t]_{ij}) \right)}_{:=\mathcal{W}} \tag{A68}$$

where the upper bound of $\mathcal{W}$ can be quantified by using Markov's inequality followed by Jensen's inequality as the following:

$$\mathcal{W} = \mathbb{P}\left( \text{sign}([\nabla\Psi(\boldsymbol{\theta}_t)]_{ij}) \neq \text{sign}([\hat{\mathbf{g}}_t]_{ij}) \right)$$
$$\leq \mathbb{P}[|[\nabla\Psi(\boldsymbol{\theta}_t)]_{ij} - [\hat{\mathbf{g}}_t]_{ij}| > [\nabla\Psi(\boldsymbol{\theta}_t)]_{ij}] \tag{A69}$$

$$\leq \frac{\mathbb{E}[[\nabla\Psi(\boldsymbol{\theta}_t)]_{ij} - [\hat{\mathbf{g}}_t]_{ij}]}{|[\nabla\Psi(\boldsymbol{\theta}_t)]_{ij}|} \tag{A70}$$

$$\leq \frac{\sqrt{\mathbb{E}[([\nabla\Psi(\boldsymbol{\theta}_t)]_{ij} - [\hat{\mathbf{g}}_t]_{ij})^2]}}{|[\nabla\Psi(\boldsymbol{\theta}_t)]_{ij}|} \tag{A71}$$

$$\overset{(A42)}{\leq} \frac{\sqrt{\mathbb{E}[([\bar{\mathbf{g}}_t]_{ij} - [\mathbf{g}_t^*]_{ij} + [\mathbf{g}_t^*]_{ij} - [\mathbf{g}_t]_{ij} + [\mathbf{g}_t]_{ij} - [\hat{\mathbf{g}}_t]_{ij})^2]}}{|[\nabla\Psi(\boldsymbol{\theta}_t)]_{ij}|} \tag{A72}$$

$$\overset{(a)}{\leq} \sqrt{3} \frac{\sqrt{\frac{(1+\lambda)\sigma_{ij}^2}{M|\mathcal{B}|} + \epsilon_{ij} + \delta_{ij}^2}}{|[\nabla\Psi(\boldsymbol{\theta}_t)]_{ij}|} \tag{A73}$$

where $(a)$ is true due to the following relations: $i$) from (A51), we have

$$\mathbb{E}[([\bar{\mathbf{g}}_t]_{ij} - [\mathbf{g}_t^*]_{ij})^2] \leq \frac{(1+\lambda)\sigma_{ij}^2}{MB}; \tag{A74}$$

$ii$) from (A49), we can get

$$\mathbb{E}[([\mathbf{g}_t]_{ij} - [\mathbf{g}_t^*]_{ij})^2] \leq \varepsilon_{ij}; \tag{A75}$$

and $iii$) from (A67), we know

$$\mathbb{E}[([\hat{\mathbf{g}}_t]_{ij} - [\mathbf{g}_t]_{ij})^2] \leq \delta_{ij}^2. \tag{A76}$$

Therefore, combining (A55) with the upper bound of $\mathcal{U}$ shown in (A66) and $\mathcal{V}$ shown in (A68)(A73), we have

$$\mathbb{E}[\Psi(\boldsymbol{\theta}_{t+1})] \leq \mathbb{E}[\Psi(\boldsymbol{\theta}_t)] - \eta_t c_l \sqrt{\frac{h(1-\beta_2)}{G^2 d}} \frac{1}{4} \mathbb{E}\|\nabla\Psi(\boldsymbol{\theta}_t)\|^2 + \eta_t c_l \sqrt{\frac{h(1-\beta_2)}{G^2 d}} \left(\varepsilon + \frac{(1+\lambda)\sigma^2}{MB}\right)$$
$$+ \eta_t c_l \sqrt{\frac{h(1-\beta_2)}{G^2 d}} \delta^2 + \eta_t c_u \sqrt{3} \sum_{i=1}^{h} \sum_{j=1}^{d_i} \sqrt{\frac{(1+\lambda)\sigma_{ij}^2}{MB} + \varepsilon_{ij} + \delta_{ij}^2} + \frac{\eta_t^2 c_u^2 \sum_{i=1}^{h} L_i}{2}. \tag{A77}$$

Note that the error vector $\boldsymbol{\chi}$ is defined as the following

$$\boldsymbol{\chi} = \begin{bmatrix} \sqrt{\frac{(1+\lambda)\sigma_{11}^2}{M|\mathcal{B}|} + \varepsilon_{11} + \delta_{11}^2} \\ \vdots \\ \sqrt{\frac{(1+\lambda)\sigma_{ij}^2}{M|\mathcal{B}|} + \varepsilon_{ij} + \delta_{ij}^2} \\ \vdots \\ \sqrt{\frac{(1+\lambda)\sigma_{hd_h}^2}{M|\mathcal{B}|} + \varepsilon_{hd_h} + \delta_{hd_h}^2} \end{bmatrix} \in \mathbb{R}^d, \tag{A78}$$

and we have

$$L = \begin{bmatrix} L_1 \\ \vdots \\ L_h \end{bmatrix} \in \mathbb{R}^h. \tag{A79}$$

Recall

$$\kappa = \frac{c_u}{c_l}. \tag{A80}$$

Rearranging the terms, we can arrive at

$$\underbrace{\sqrt{\frac{h(1-\beta_2)}{G^2 d}} \frac{1}{4}}_{:=C} \left(\|\nabla\Psi(\boldsymbol{\theta}_t)\|^2\right) \leq \frac{\mathbb{E}[\Psi(\boldsymbol{\theta}_t)] - \mathbb{E}[\Psi(\boldsymbol{\theta}_{t+1})]}{\eta_t c_l} + 4C\delta^2 + 2C\left(\varepsilon + \frac{(1+\lambda)\sigma^2}{MB}\right)$$
$$+ \sqrt{3}\kappa\|\boldsymbol{\chi}\|_1 + \frac{\eta_t c_u \kappa \|L\|_1}{2}. \tag{A81}$$

Applying the telescoping sum over $t = 1, \ldots, T$, we have

$$\frac{1}{T} \sum_{t=1}^{T} \mathbb{E}\|\nabla_{\boldsymbol{\theta}}\Psi(\boldsymbol{\theta}_t)\|^2 \leq \frac{\mathbb{E}[\Psi(\boldsymbol{\theta}_1)] - \mathbb{E}[\Psi(\boldsymbol{\theta}_{T+1})]}{\eta_t c_l CT} + 2\left(\varepsilon + \frac{(1+\lambda)\sigma^2}{MB}\right) + 4\delta^2$$
$$+ \frac{\kappa\sqrt{3}}{C}\|\boldsymbol{\chi}\|_1 + \frac{\eta_t c_u \kappa \|L\|_1}{2C}. \tag{A82}$$

$\square$

## 4 ADDITIONAL EXPERIMENTS

### 4.1 DISCUSSION ON CYCLIC LEARNING RATE

It was shown in (Wong et al., 2020) that the use of a cyclic learning rate (CLR) trick can further accelerate the Fast AT algorithm in the small-batch setting (Wong et al., 2020). In Figure A1, we present the performance of Fast AT with CLR versus batch sizes. We observe that when CLR meets the large-batch setting, it becomes significantly worse than its performance in the small-batch setting. The reason is that CLR requires a certain number of iterations to proceed with the cyclic schedule. However, the use of large data batch only results in a small amount of iterations by fixing the number of epochs.

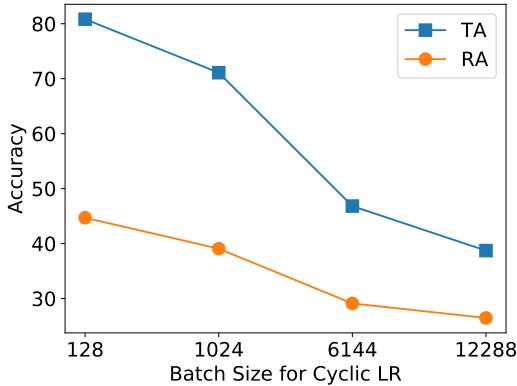

Figure A1: TA/RA of Fast AT with CLR versus batch sizes.

### 4.2 TRAINING DETAILS

CIFAR-10 AT and Fast AT experiments are conducted at a single computing node with 16-core CPU, 128GB RAM and 1 Nvidia P100 GPU. The training epoch is 100 by calling for the momentum SGD optimizer. The weight decay and momentum parameters are set to 0.0005 and 0.9. The initial learning rate is set with 0.05 (tuned over $\{0.005, 0.01, 0.05, 0.1\}$), which is decayed by $\times 1/10$ at the training epoch 70, 85 and 95, respectively.

CIFAR-10 DAT experiments are conducted at $\{1, 6, 12, 18\}$ computing nodes with 16-core CPU, 128GB RAM and 1 Nvidia P100 GPU. The training epoch is 100 by calling for the LAMB optimizer. The weight decay is set to 0.0005. $\beta_1$ and $\beta_2$ are set to 0.9 and 0.999. The initial learning rate is tuned over $\{0.01, 0.05, 0.1, 0.2\}$, which is decayed by $\times 1/10$ at the training epoch 70, 85 and 95, respectively. To execute algorithms with the initial learning rate $\eta_1$ greater than 0.1, we choose the model weights after 10-epoch warm-up as its initialization for DAT, where each warm-up epoch $k$ uses the linearly increased learning rate $(k/10)\eta_1$.

ImageNet AT and Fast AT experiments are conducted at a single computing node with dual 22-core CPU, 512GB RAM and 6 Nvidia V100 GPUs. The training epoch is 30 by calling for the momentum SGD optimizer. The weight decay and momentum parameters are set to 0.0001 and 0.9. The initial learning rate is set to 0.1 (tuned over $\{0.01, 0.05, 0.1, 0.2\}$), which is decayed by $\times 1/10$ at the training epoch 20, 25, 28, respectively.

ImageNet DAT experiments are conducted at $\{1, 3, 6\}$ computing nodes with dual 22-core CPU, 512GB RAM and 6 Nvidia V100 GPUs. The training epoch is 30 by calling for the LAMB optimizer. The weight decay is set to 0.0001. $\beta_1$ and $\beta_2$ are set to 0.9 and 0.999. The initial learning rate is tuned over $\{0.01, 0.05, 0.1, 0.2, 0.4\}$, which is decayed by $\times 1/10$ at the training epoch 20, 25, 28, respectively. To execute algorithms with the initial learning rate $\eta_1$ greater than 0.2, we choose the model weights after 5-epoch warm-up as its initialization for DAT, where each warm-up epoch $k$ uses the linearly increased learning rate $(k/5)\eta_1$.

**Empirical model convergence.** In Figure A2, we present the training accuracy and the loss value of DAT-PGD. As we can see, our proposal converges well within 100 and 30 epochs in the setting of (CIFAR-10, ResNet-18) and (ImageNet, ResNet-50), respectively

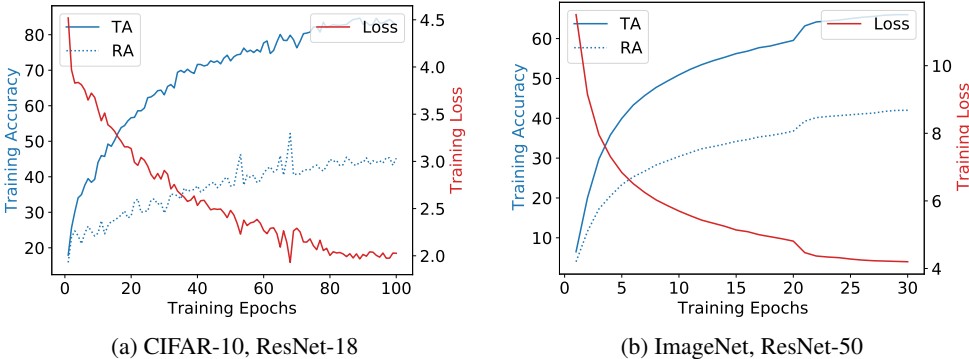

(a) CIFAR-10, ResNet-18          (b) ImageNet, ResNet-50

Figure A2: Training accuracy and objective value (loss) of DAT-PGD against training epochs. (a) DAT-PGD for (CIFAR-10, ResNet-18) using $6 \times 1$ computing configuration and $6 \times 2048$ batch size. (b) DAT-PGD for (ImageNet, ResNet-50) using $6 \times 6$ computing configuration and $6 \times 512$ batch size.

**Tuning LALR hyperparameter $c_u$.** We also evaluate the sensitivity of the performance of DAT to the choice of the hyperparameter $c_u$ in LALR. In Table A1, we fix $c_l = 0$ (this is a natural choice) but varies $c_u \in \{8, 9, 10, 11, 12\}$ when DAT-FGSM is executed under CIFAR-10 using $18x2048$ batch size, where $c_u = 10$ is our default choice. As we can see, both RA and TA are not quite sensitive to $c_u$ and the default choice yields the RA-best model (in spite of minor improvement).

Table A1: TA/RA of DAT-FGSM under (CIFAR-10, ResNet-18) using $18x2048$ batch size versus different choices of LALR hyperparameter $c_u$.

| Value of $c_u$ | TA (%) | RA (%) |
|---|---|---|
| $c_u = 8$ | 73.57 | 38.19 |
| $c_u = 9$ | 73.72 | 38.00 |
| $c_u = 10$ | 73.42 | 38.55 |
| $c_u = 11$ | 73.75 | 38.18 |
| $c_u = 12$ | 73,63 | 37.87 |

### 4.3 OVERALL PERFORMANCE OF (CIFAR-10, RESNET-50)

In Table A2, we observe that in the large-batch setting, the proposed DAT-PGD and DAT-FGSM algorithms outperform the baseline algorithm DAT-PGD w/o LALR, and result in competitive performance to AT and Fast AT, which call for more iterations by using a smaller batch size.

Table A2: Overall performance of DAT (in gray color), compared with baselines, in TA (%), RA (%), communication time per epoch (seconds), and total training time (including communication time) per epoch (in seconds). For brevity, '$p \times q$' represents '# nodes $\times$ # GPUs per node', 'Comm.' represents communication cost, and 'Tr. Time' represents training time.

| Method | CIFAR-10, ResNet-50 | | | | | |
|---|---|---|---|---|---|---|
| | $p \times q$ | Batch size | TA (%) | RA (%) | Comm. per epoch (s) | Tr. Time per epoch (s) |
| AT | $1 \times 1$ | 256 | 85.94 | 43.06 | NA | 894 |
| Fast AT | $1 \times 1$ | 256 | 75.28 | 40.48 | NA | 288 |
| DAT-PGD w/o LALR | $6 \times 1$ | $6 \times 256$ | 74.45 | 33.35 | 68 | 236 |
| DAT-PGD | $6 \times 1$ | $6 \times 256$ | 84.79 | 42.16 | 68 | 236 |
| DAT-FGSM | $6 \times 1$ | $6 \times 256$ | 75.72 | 40.09 | 68 | 116 |

## 4.4 ROBUSTNESS AGAINST PGD AND C&W ATTACKS

In Figure A3, we evaluate the adversarial robustness of ResNet-18 at CIFAR-10 learned by DAT-PGD and DAT-FGSM against PGD attacks of different steps and perturbation sizes (namely, values of $\epsilon$). We consistently observe that DAT matches robust accuracies of standard AT even against PGD attacks at different values of $\epsilon$ and steps. Specifically, DAT has slightly smaller RA than AT when facing weak PGD attacks with $\epsilon$ less than $(5/255)$ and steps less than 5. Moreover, although DAT-FGSM has the worst RA against weak PGD attacks (which reduces to TA at $\epsilon = 0$), it outperforms other methods when the attacks become stronger in CIFAR-10 experiments. In Figure A4, we present the additional robust accuracies against C&W attacks (Carlini & Wagner, 2017) of different perturbation sizes. As we can see, the results are consistent with the aforementioned ones against PGD attacks.

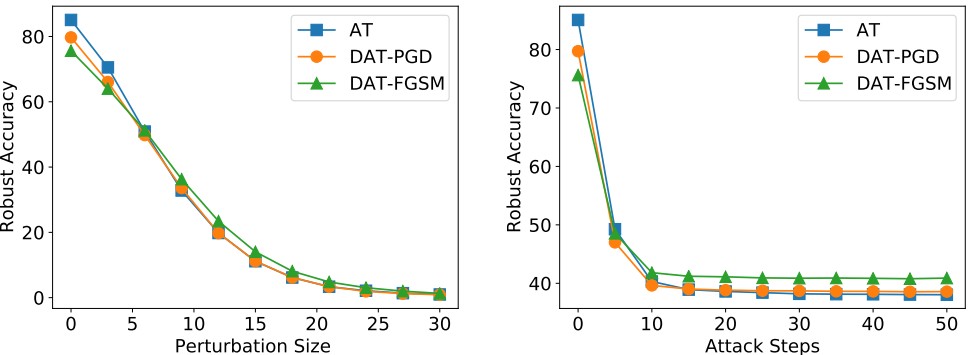

Figure A3: RA against different PGD attacks for the model trained by DAT-PGD, DAT-FGSM, and AT under (CIFAR-10, ResNet-18). (Left) RA against PGD attacks with different perturbation sizes (over the divisor 255). (Right) RA against PGD attacks with different steps.

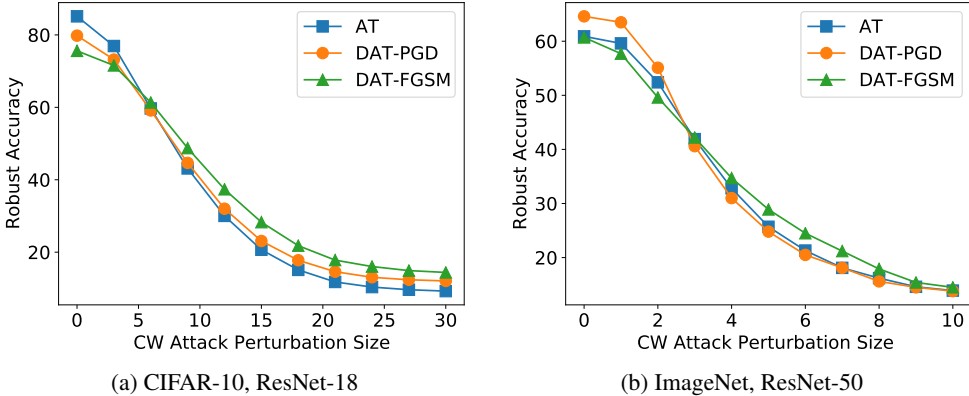

(a) CIFAR-10, ResNet-18

(b) ImageNet, ResNet-50

Figure A4: RA against different C&W attacks for the model trained by DAT-PGD, DAT-FGSM, and AT under the setting (CIFAR-10, ResNet-18) and (ImageNet, ResNet-50), respectively. Here for ease of C&W attack generation at ImageNet, we randomly select 1000 test ImageNet images (1 image per class) to generate C&W attacks.

### 4.5 DAT FROM PRE-TRAINING TO FINE-TUNING

In Figure A5, we investigate if a DAT pre-trained model (ResNet-50) over a source dataset (ImageNet) can offer a fast fine-tuning to a down-stream target dataset (CIFAR-10). Compared with the direct application of DAT to the target dataset (without pre-training), the pre-training enables a fast adaption to the down-stream CIFAR-10 task in both TA and RA within just 5 epochs.

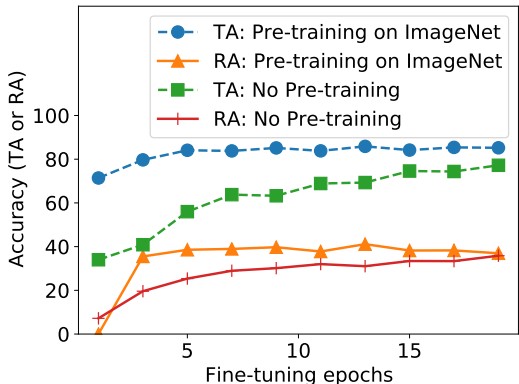

Figure A5: Fine-tuning ResNet-50 (pre-trained on ImageNet) under CIFAR-10. Here DAT-PGD is used for both pre-training and fine-tuning at 6 nodes with batch size $6 \times 128$.

### 4.6 QUANTIZATION

In Table A3, we present the performance of DAT by making use of gradient quantization. Two quantization scenarios are covered: 1) quantization is conducted at each worker (Step 7 of Algorithm A1), and 2) quantization is conducted at both worker and server sides (Step 7 and 10 of Algorithm A1). As we can see, when the number of bits is reduced from 32 to 8, the communication cost and the amount of transmitted data is saved by 2 and 4 times, respectively. Although the use of gradient quantization introduces a performance loss to some extent, the resulting TA and RA are still comparable to the best 32-bit case. In the worst case of CIFAR-10 (8-bit 2-sided quantization), TA drops $0.91\%$ and $6.33\%$ for DAT-PGD and DAT-FGSM, respectively. And RA drops $4.73\%$ and $5.22\%$, respectively. However, 8-bit 2-sided quantization transmitted the least amount of data per iteration.

To further reduce communication cost, we also conduct DAT at a HPC cluster. The computing nodes of the cluster are connected with InfiniBand (IB) and PCIe Gen4 switch. To compare with results in Table 1, we use 6 of 57 nodes of the cluster. Each node has 6 Nvidia V100s which are interconnected with NVLink. We use Nvida NCCL as communication backend. In Table A4, we present the performance of DAT for ImageNet, ResNet-50 with use of HPC compared to standard (non-HPC) distributed system. As we can see, the communication cost is largely alleviated, and thus the total training time is further reduced.

In Table A5, we conduct an additional experiment by integrating a centralized method with gradient quantization operation on CIFAR-10 under the batch size $2048$ and $6 \times 2048$, respectively. We specify the centralized method as Fast AT with LALR, where LALR is introduced to improve the scalability of Fast AT under the larger batch size $6 \times 2048$. Due to the centralized implementation, we only need 1-sided gradient quantization (namely, no server-worker communication is involved). As we can see, when the batch size $2048$ is used, Fast AT w/ LALR performs as well as Fast AT even at the presence of 8-bit gradient quantization. On the other hand, as the larger batch size $6 \times 2048$ is used, Fast AT w/ LALR can still preserve the performance at the absence of gradient quantization. By contrast, Fast AT w/ LALR at the presence of quantization encounters $6.05\%$ TA drop. This suggests that even in the non-DAT setting, 8-bit gradient quantization hurts the performance as the batch size becomes large. Thus, in DAT it is not surprising that 8-bit quantized gradients could cause a non-trivial accuracy drop, particularly for using 2-sided gradient quantization and a much larger data batch size ($\geq 18x2048$ on CIFAR-10). One possible reason is that the quantization error cannot easily be mitigated as the number of iterations decreases (due to increased batch size under a fixed number of epochs).

Table A3: Effect of gradient quantization on the performance of DAT for various numbers of bits. The training settings of (CIFAR-10, ResNet-18) and (ImageNet, ResNet-50) are consistent with those in Table 1.

| Method | CIFAR-10, ResNet-18 | | | | |
|---|---|---|---|---|---|
| | # bits | TA (%) | RA (%) | Comm. per epoch (s) | Data transmitted per iteration (MB) |
| DAT-PGD | 32 | 80.38 | 38.94 | 8.5 | 1278 |
| DAT-PGD | 16 | 79.38 | 38.32 | 8.3 | 639 |
| DAT-PGD | 8 | 78.18 | 37.34 | 4.3 | 320 |
| DAT-PGD | 8 (2-sided) | 78.86 | 34.2 | 5.0 | 107 |
| DAT-FGSM | 32 | 75.58 | 40.92 | 8.5 | 1278 |
| DAT-FGSM | 16 | 75.74 | 40.86 | 8.3 | 639 |
| DAT-FGSM | 8 | 72.48 | 38.98 | 4.3 | 320 |
| DAT-FGSM | 8 (2-sided) | 69.26 | 35.34 | 5.0 | 107 |
| Method | ImageNet, ResNet-50 | | | | |
| | # bits | TA (%) | RA (%) | Comm. per epoch (s) | Data transmitted per iteration (MB) |
| DAT-PGD | 32 | 63.75 | 38.45 | 898 | 2924 |
| DAT-PGD | 16 | 61.77 | 38.40 | 850 | 1462 |
| DAT-PGD | 8 | 56.53 | 37.90 | 592 | 731 |
| DAT-PGD | 8 (2-sided) | 53.09 | 34.59 | 1091 | 244 |
| DAT-FGSM | 32 | 58.32 | 41.48 | 859 | 2924 |
| DAT-FGSM | 16 | 54.71 | 39.29 | 849 | 1462 |
| DAT-FGSM | 8 | 50.11 | 36.38 | 594 | 731 |
| DAT-FGSM | 8 (2-sided) | 48.27 | 33.20 | 1013 | 244 |

Table A4: Comparison to training over a high performance computing (HPC) cluster of nodes

| Method | ImageNet, ResNet-50 | | | | |
|---|---|---|---|---|---|
| | # bits | TA (%) | RA (%) | Comm. per epoch (s) | Tr. time per epoch (s) |
| DAT-PGD | 32 | 63.75 | 38.45 | 898 | 1960 |
| DAT-FGSM | 32 | 58.32 | 41.48 | 859 | 1109 |
| DAT-PGD (HPC) | 32 | 63.43 | 38.55 | 15 | 1074 |
| DAT-FGSM (HPC) | 32 | 57.60 | 41.70 | 15 | 310 |

Table A5: Effect of 8-bit quantization on centralized robust training Fast AT w/ LALR.

| Method | CIFAR-10, ResNet-18 | | | |
|---|---|---|---|---|
| | 8-bit quantization (s) | Batch size | TA (%) | RA (%) |
| Fast AT | No | 2048 | 81.58 | 38.34 |
| Fast AT w/ LALR | Yes | 2048 | 80.66 | 38.60 |
| Fast AT w/ LALR | No | 6 x 2048 | 80.08 | 38.51 |
| Fast AT w/ LALR | Yes | 6 x 2048 | 75.53 | 38.45 |

