# OpenReview forum: "Distributed Adversarial Training to Robustify Deep Neural Networks at Scale"
_ICLR.cc/2021/Conference — Reject_

### Official Review · AnonReviewer4 · 2020-10-26
**I have some concerns about differences from existing work and the experiments**

**Rating:** 4
**Confidence:** 4

**Review:**

Hello authors,

Thank you for the submission.  I found it to be well-written and I enjoyed reading it.  The generic description of DAT given in Equation 2 is nice.  However, there are some concerns on my end that I hope you can help me alleviate that are keeping me from giving the paper a positive review.  I have an open mind and can be convinced.

1. Why is the solution of DAT not obvious?

The authors claim that "the direct solution [for scaling AT] of distributing the data batch across multiple machines may not work", but I cannot understand why when Algorithm 1 looks like exactly that.  The adversarial samples are generated on each machine at the beginning of each iteration so as to avoid computing them on the server at the beginning of each iteration.  To me, this is an unsurprising and obvious optimization.  Other than that minor variation, this looks exactly like the standard parameter server approach.

2. Are the experiments really fair here?

The most clear claim appears to be from Table 1 and is mentioned in the introduction, where using DAT-PGD with 6 nodes gives a 3x speedup over AT while preserving standard test accuracy and almost preserving robust accuracy.  However, compare with when LALR is *not* used: then, the accuracy is significantly degraded.  So a natural question that I would like to understand is what LALR does when used with the original AT algorithm.  In fact, to get a clear apples-to-apples comparison, I would think that it would be best to use AT with a 6*512 batch size, plus LALR.  Basically, what I mean is that I am not sure if the preservation of accuracy is entirely due to LALR.  If it is, this implies that LALR can be used off-the-shelf with AT in a *non*-distributed setting with larger batch size, and that could result in a situation where DAT-PGD + LALR significantly underperforms AT + LALR---which would contradict the result that DAT is able to preserve accuracy, instead giving the result that LALR improves accuracy.

Alternately, I might expect that DAT-PGD without LALR but with a batch size of 512/6 on each of the 6 nodes (giving the same as the original batch size) would give comparable performance.

I hope the gist of my comment is clear here: it seems like the comparison is not really a fair comparison.  I think it muddles improvements from DAT and improvements from LALR, and makes it hard for the reader to understand what is really going on.

Some additional small things:

 - In (1) boldface y is used, but in (2), non-boldface y is used.  Perhaps these should be made the same?
 - "Server" instead of "Sever" in Algorithm 1
 - In Section 4 there seems to be redundancy: "see the meta-form of DAT in Algorithm 1 and details in Algorithm A1. " but I believe it is meant that both references should point at the same thing.
 - In Section 4, it is written that "DAT takes a M times fewer iteration than AT".  (Also there are some grammar issues in that sentence.)  It may be clearer to say "M fewer gradient updates on the server".
 - In Section 4, it is claimed that the computation time of DAT with iterative PGD is K times more than that of DAT with FGSM.  I know that I am being pedantic but this is not correct---to make it correct, the computation time of *the adversarial perturbation generation step* is K times more, not the overall computational time.
 - Use \citet not \citep at the end of the sentence starting with "Indeed, we will show ..." at the top of page 5.
 - What is the unit for training time in Table 1?  Does that include communication time?

Overall, I am open to discuss my thoughts, but in my current understanding, I am not able to make a strong enough case for DAT to be published at ICLR.

## Post-rebuttal comments

I am impressed at the amount of time the authors have spent trying to clarify the different concerns.  But unfortunately my concerns are not fully addressed, and also a new concern arises: if this much space had to be spent clarifying different confusions of the reviewers, I think the paper could do with a complete overhaul.  I would suggest that the authors take into account the general confusions that arose in this review process and rewrite the paper such that those particular confusions are alleviated.  For instance, I struggled for clarity on what makes this contribution non-trivial, whether the contributions are primarily theoretical or primarily empirical (and how I should understand the balance between the two), and the fairness of the empirical comparison.  Rewriting the paper such that those three concerns---and the others pointed out by the other reviewers---are discussed clearly would be a significant improvement.

---

> ### Author Response · Authors · 2020-11-18
> **Response to Reviewer 4 - Part 2**
>
> ### **Are the experiments really fair here?**
>
> Thanks for raising these questions. We understood your concerns on (1) AT + LALR using large batch size vs. DAT-PGD ( + LALR), and (2) AT vs. DAT-PGD w/o LALR using small batch size. However, there might exist some misunderstandings on why DAT is needed as an alternative to its centralized counterpart. We need DAT since **centralized methods** (e.g., the suggested AT + LALR or AT) become **infeasible**, e.g., in the following scenarios.  First, if (private) data is required for distributed storage, then the centralized method cannot be used. Besides, if the data batch exceeds the storage capacity of a single machine, then the distributed method is desired. In our case (and many practical scenarios), the number of GPUs provided by a single machine is limited (the maximum number is $6$ given our accessed resources), and thus the largest batch size that the centralized method (e.g., the suggested AT + LALR) can use is upper bounded, e.g., $512$ rather than $6 \times 512$ in our case. That is, AT + LALR with $6 \times 512$ batch size is not a feasible centralized baseline for us since our single machine cannot support such a large batch. To enable comparison under large batch size, we thus provided the distributed implementation of AT and Fast AT, given by DAT-PGD w/o LALR and DAT-FGSM w/o LALR, respectively.
>
> On the other hand, if a large data batch is not necessary (i.e., no need for training acceleration), then the centralized method (AT + LALR or AT) could be executed given the condition that a single machine can afford the batch size. It is expected that the *centralized* solution can outperform the *distributed* solution as the former is free of machine synchronization and communication. Following the reviewer’s suggestion, we conduct additional experiments on CIFAR-10 when both centralized and distributed methods are eligible in comparison under the same batch size ($2048$).
>
> | Method       | Machines x batch size per machine | TA | RA |
> |--------------|-----------------------------------|----|----|
> | **DAT + LALR**   | $6\times341$                             |$83.48$|$39.76$|
> |**AT + LALR**   | $2048$                              |$83.56$|$39.84$|
> | **DAT w/o LALR** |$6\times341$                             |$82.42$|$38.41$|
> | **AT w/o LALR**  | $2048$                              |$82.94$|$38.54$|
>
> As we can see, the distributed methods (DAT-PGD or DAT-PGD w/o LALR) can yield comparable performance (with slight degradation) to the centralized methods (AT + LALR or AT). And the performance under the small batch size is not sensitive to LALR.
>
> All in all, we hope that the reviewer did not mix up the centralized and distributed baselines. If the batch size is small (applicable to centralized cases), then the centralized AT outperforms its distributed counterpart, and the performance does not rely on LALR. However, if the batch size is large (inapplicable to centralized cases), then DAT + LALR outperforms DAT (namely, LALR matters).
>
> ### **Other minor comments:**
>
> Thanks for the careful reading. We will correct typos and grammar issues, and update the paper. In the original caption of Table 1, we have mentioned that the unit of training time per epoch in seconds and the total training time includes the communication time.
>
> We hope our response has mostly addressed your concerns, and we hope it highlights our efforts in making a thorough study of distributed adversarial training when centralized training becomes inapplicable. We are glad to continue a discussion to address any other questions you may have.

---

> ### Author Response · Authors · 2020-11-18
> **Response to Reviewer 4 - Part 1**
>
> Thank you for the detailed and very insightful comments. We’re very glad you enjoyed reading our work, and likewise, we found the set of perceptive questions you raised very insightful, pushing us to think of a tighter explanation on the need and novelties of DAT. We address the comments and feedback you provided below.
>
>
> ###  **Why is the solution of DAT (Algorithm 1) not obvious?**
>  Algorithm 1 is only a meta-form of our detailed Algorithm A1 (Appendix 1), and follows the generic parameter-server setting. The generation of adversarial examples is not the main bottleneck across distributed machines as the reviewer pointed out. By contrast, the support of large data batches (including labeled and unlabeled data) across machines is the main challenge. Although either of the standalone techniques, adversarial training, gradient quantization, and large-batch standard training, have been established separately, justifying this seemingly straightforward combination **actually works** is indeed **not trivial**.  In fact, these standalone techniques for the first time are proven and shown effective when used jointly in DAT through our theoretical analysis and empirical results. Thus, we do not think that our proposal is an ‘obvious’ solution in both theory and practice. Please see the details below.
>
> * In theory, the incorporation of layerwise adaptive learning rate (LALR) improves DAT’s scalability but makes its convergence rate analysis far from trivial. The fundamental challenge lies in the nonlinear coupling between the biased gradient estimate resulting from LALR and the additional error generated from alternating updates in min-max optimization. We prove in Theorem 1 that even in the case where the gradient estimate is a function of the AT variables, the estimate bias resulting from the layerwise normalization can still be compensated by increasing the batch-size so that the convergence rate of DAT is preserved as the original large batch training algorithm and the gradient estimate error is shrunk as the number of computing nodes increases (achieves a linear speedup w.r.t. M computing nodes). To the best of our knowledge, we for the first time establish the convergence rate analysis for large-batch min-max optimization. This is an exciting optimization result.
> * In practice, as shown in Table 1, the direct distributed implementation of AT (i.e., DAT-PGD w/o LALR) and the direct distributed solution of Fast-AT (i.e., DAT-FGSM w/o LALR) cannot scale to large batch sizes. Thus, we conducted extensive experiments to show the effectiveness of DAT. For example, the performance improvement of our approach over the large-batch SGD (LSGD) baseline (Xie et al. 2019) has been shown in Figure 1 under different distributed learning settings. We also show the scalability of DAT lies in not only the large-batch computation setting but also the pre-training + fine-tuning setting (Figure 3) and the gradient quantization-enabled computing configuration  (AllReduce, parameter-server, and HPC setups; see Sec. 5 & Appendix 4.6).
>
> In summary, our formulation, algorithm, theory, and experiments make the first thorough study on distributed robust training. We strongly believe that the ‘obvious’ meta algorithm does not imply ‘trivial’ contributions as we explained above.

---

> > ### Comment · AnonReviewer4 · 2020-11-24
> > **Response to Response to Reviewer 4 - Part 1**
> >
> > Thanks for the response.  I will try and be concise.
> >
> > The amount of space in the rebuttal spent justifying that DAT is non-trivial---both to me and to other reviewers---suggests strongly to me that the paper needs to be overhauled to be clearer about the contributions.  If the primary contribution is theoretical, more than ~a page should be spent discussing those results.  If the primary contribution is empirical, more convincing arguments will be required for why the empirical contribution is nontrivial.  Though the theory may be nontrivial, combining adversarial training, gradient quantization, and large-batch standard training does not appear to be difficult in practice.
> >
> > > In practice, as shown in Table 1, the direct distributed implementation of AT (i.e., DAT-PGD w/o LALR) and the direct distributed solution of Fast-AT (i.e., DAT-FGSM w/o LALR) cannot scale to large batch sizes.
> >
> > I don't follow the justification for this, and I don't see any particular reason why this shouldn't work other than implementational inefficiency.  If the AT and Fast-AT authors' code does not work for large batch sizes, but the GPU has enough memory to be able to work with large batch sizes (which it clearly does as all DAT implementations work), this is not a sufficient reason to not include an apples-to-apples comparison.  The listed contributions of DAT are to combine three pieces: adversarial training, gradient quantization, and large-batch standard training.  I can think of no reason why large-batch standard training should not also work for AT and Fast AT.
> >
> > Thank you to the authors for your comments, but in this case I will leave my assessment as-is.

---

> > > ### Author Response · Authors · 2020-11-24
> > > **Further response**
> > >
> > > Thanks for R#4's further comments. We feel that the reviewer might misunderstand our points.
> > >
> > > >the GPU has enough memory to be able to work with large batch sizes (which it clearly does as all DAT implementations work).
> > >
> > > This is true for DAT since multiple distributed machines are used, but each machine has a limited number of GPUs. Thus, the centralized methods AT and Fast AT (at a single machine) without calling for distributed implementation are not directly comparable to the distributed setting in a large-batch setting. Thus, there exist **two ways** that we can compare:
> > >
> > > **1) Fixing the large batch size same as DAT, and building the distributed versions of AT and Fast AT, which thus correspond to DAT-PGD w/o LALR and DAT-FGSM w/o LALR, respectively.**  As we can see, their resulted TA/RA is worse than the proposed DAT in the large-batch case. Thus, we claimed that they cannot scale to large batch settings. We also would like to point out that in the large-batch case, the considered distributed baselines DAT-PGD w/o LALR and DAT-FGSM w/o LALR are fair. To further clarify this point, we exactly follow the ImageNet setting of Fast-AT (Wong et al., 2020), which consists of three training phases using different batch sizes (due to image resizing), 512 for phase 1, 224 for phase 2, and 128 for phase 3. And we conduct new experiments by increasing the batch size to the maximum value that each training phase supports, given by 4096 for phase 1, 1300 for phase 2, and 384 for phase 3. As a result, the direct distributed implementation of Fast AT (Wong et al., 2020)  yields **43.67% (TA)** and **28.10% (RA)**, which are much worse than the performance of DAT-FGSM w/o LALR. Thus, our considered DAT-FGSM w/o LALR actually gives a reasonable distributed baseline of Fast AT.
> > >
> > >
> > > **2) Reducing the batch size such that a single machine can easily afford.** Please refer to Part-2 Response https://openreview.net/forum?id=kmBFHJ5pr0o&noteId=uxRhj6f3bZW
> > > As we can see, when the batch size is not large,  all methods, AT, Fast AT, and DAT perform well, yielding very similar performances. However, training will smaller batch size is less efficient as it will increase the overall number of iterations.
> > >
> > > >The listed contributions of DAT are to combine three pieces: adversarial training, gradient quantization, and large-batch standard training. I can think of no reason why large-batch standard training should not also work for AT and Fast AT.
> > >
> > > As we explained earlier and in the next response, if the centralized method is allowed (namely, having access to a sufficient number of GPUs), then the large-batch standard training will work for AT and Fast AT since they become the centralized implementations of DAT.  However, we aim to show when centralized training with large-batch is infeasible, DAT can scale up adversarial training across distributed machines,  e.g.,  using $6 \times 512$ for ImageNet across 6 machines. This is not the regime that the centralized case can achieve. Even in the distributed setting, although combining three pieces (adversarial training, gradient quantization, and large-batch standard training) works (as Reviewer believed and DAT showed), it still requires careful empirical studies in generalizability (supervision and semi-supervision, pre-training and fine-tuning, PGD and FGSM attack generation) and scalability (large-batch learning and quantization-aware communication). Thus, we do believe that our contribution lies in both theory and practice.

---

### Official Review · AnonReviewer3 · 2020-10-28
**The paper proposes a novel and comprehensive distributed learning method for speeding up adversarial training with multiple computing nodes.**

**Rating:** 8
**Confidence:** 4

**Review:**

Adversarial training is a principled approach towards robust neural networks against adversarial attacks, but it is extremely computing intensive. This work tackles the problem by leveraging the general distributed training method, and addressing the problems of direct application by several effective innovations.

Strength:

+ The proposed method is practical.

+ The proposed DAT can deal with both labeled data (supervised learning) and partial unlabeled data (semi-supervised).

+ I like the idea to use gradient quantization/compression.

+ Make theorectical conribution by convergence analysis for DAT with LALR and gradient quantization.

Additional comments and questions:

1. One conclusion is the paper can speed up by 3 times with 6 times resource. Please comment on what's the typical speedup of distributed training with n times of resources. Do you think you can further speedup?

2. In formula (2), additional regularization term by lamba is used, making it different from the direct generation of (1) into multiple workers. Why we must introduce the regularization term in DAT?

3. Please provide more details about how to measure communication time please? any profiler used?

4. In table 2, additional unlabeled data can improve accuray. Why?

5. Which deep learning framework is used to support gradient quantization?

---

> ### Author Response · Authors · 2020-11-18
> **Response to Reviewer 3 - Part 2**
>
> ### **Why additional unlabeled data can improve accuracy?**
>
>  In the supervised setting, it has been shown that there is a tradeoff between robustness and accuracy [2]. However, it was recently shown in [3,4] that both accuracy and robustness can further be improved by leveraging unlabeled data; e.g., Table 1 of [3]. Our empirical results in DAT by leveraging unlabeled data echoes the advantage of unlabeled data.
>
>   [2] Tsipras, Dimitris, et al. "Robustness may be at odds with accuracy." arXiv preprint arXiv:1805.12152 (2018).
>
>   [3] Carmon, Yair, et al. "Unlabeled data improves adversarial robustness." Advances in Neural Information Processing Systems. 2019.
>
>   [4] Alayrac, Jean-Baptiste, et al. "Are Labels Required for Improving Adversarial Robustness?." Advances in Neural Information Processing Systems. 2019.
>
> ### **Which deep learning framework used to support quantization?**
>
> We used `torch` and `torch.distributed` as learning framework, over which we built a customized gradient quantization function following Eq. A5 and A6.

---

> ### Author Response · Authors · 2020-11-18
> **Response to Reviewer 3 - Part 1**
>
> Thank you for seeing our extensive efforts to make a novel and comprehensive study on distributed adversarial training. And likewise, we found the set of perceptive questions you raised in your feedback very insightful, pushing us to think of how to further improve our submission. We address the comments and feedback you provided below.
>
> ### **Typical speedup of distributed training with n times of resources, and further speed-up?**
>
> Very good question! We will improve our original speed-up analysis to make it clearer.
>
> In an *ideal* case, DAT can achieve a linear speed-up. However, in practice, this is not the case because of the communication cost (main factor) and other minor factors such as rounding effect of batch size, and synchronizing cost across multiple machines. Let us take ImageNet experiments in Table 1 as an example. When comparing the computation time of DAT-PGD with that of AT, the speed-up (by excluding the communication cost) is given by $(6022)/(1960-898) = 5.67$, consistent with the ideal computation gain using $6$x larger data batch size in DAT-PGD. In CIFAR-10 experiments, the rounding effect of batch size may also play a role in affecting the practical speed-up. Let $D$ be the size of the dataset, $b$ be the batch size per node, $n$ be the number of computing nodes, and $[\cdot]$ be the ceiling function. Then, the theoretical speed-up that can be achieved by DAT versus AT  is given by$\frac{ [D/b]}{ [D/(bn)] }$. In Table 1, when comparing  DAT-PGD with AT under CIFAR-10, we have $ [D/b] = 25$ and $ [D/(bn)] = 2$, and $\frac{ [D/b]}{ [D/(bn)] } = 12.5$. This is consistent with the actual computing speed-up measured by the computation time, $218/18.6 = 11.7$.
>
> In summary, given $n$ times resources, it will lead to $\frac{ [D/b] c1 }{ [D/(bn)] (c1 + c2) }$ training speedup, where $[D/b]$ and $[D/(bn)] $ are the numbers of iterations required per epoch in centralized training and DAT respectively, and $c1$ and $c2$  are the per-iteration computation cost and communication cost respectively. Thus, we believe that reducing the communication cost (namely, $c2$) is the main approach to further speed up, like using HPC instead of AllReduce communication protocol.
>
> ### **In formula (2), why $\lambda$ in DAT?**
>
> Thanks for the question. The introduction of $\lambda$ makes our proposed framework more general so that it can cover different versions of DAT that we considered and implemented in experiments, e.g., supervised DAT  and semi-supervised DAT; see discussion after Eq. (2). Specifically, if $\lambda = 0$ and $\phi = \ell$, then (2) becomes the direct generalization of (1), commonly-used for supervised adversarial training. By contrast, when unlabeled data are incorporated for semi-supervised adversarial training, then it is common to set $\lambda > 0$ (corresponding to a standard CE loss over labeled data) and choose $\phi$ as a label-free robustness regularization that can take into account unlabeled data (e.g., Eq. (3)). Therefore, we introduce $\lambda$ in (2) toward a unified DAT framework.
>
> ### **How to measure communication time please? any profiler used?**
> We use `torch.distributed` package with `gloo` and `nccl` as communication backend[1].
> We then measure the time of required worker-server communications per epoch. We use the `time` module to measure communication time with collective `torch.distributed.barrier` to synchronize all processes on each node. There is no profiler used so far.
> In our experiments, we consider three different communication protocols, AllReduce (with one-sided quantization), parameter-server (with double quantization), and HPC setting (without quantization), which all fall into the general framework listed in Algorithm 1. Note that in AllReduce, every node performs as a server, and thus it does not need server-to-worker communication at Step 8 of Algorithm 1. The main paper focuses on the AllReduce setting. In Appendix 4.6, we have compared AllReduce with the general parameter-server and HPC setting.
>
> [1]https://pytorch.org/docs/stable/distributed.html

---

### Official Review · AnonReviewer1 · 2020-10-28
**Official Blind Review #1**

**Rating:** 5
**Confidence:** 4

**Review:**

This work introduces a framework, called DAT, to scale out adversarial training to distributed settings. DAT combines three techniques: one-shot fast gradient sign method for more efficient inner maximization, gradient quantization for reduced communication volume, and layer-wise adaptive learning rate optimizer for large-batch training.  Equipped with the proposed techniques, the authors obtain promising speedups to perform adversarial training against ResNet18 and ResNet50 on CIFAR-10 and ImageNet with multi-node and multi-GPU, while achieving comparable (roughly under 2% difference) accuracy.


Pros:
- By combing all the techniques, the proposed framework yields promising results for distributed adversarial training.
- The paper shows that sparse gradients and large-batch training scheme also apply to adversarial training and provide some theoretical analysis to show the convergence rate of the composite of these schemes.

Cons:
- The technical contribution of the work seems to be limited. All techniques used are from existing works. Therefore, the main contribution is on applying these techniques in the context of adversarial training.
- Several optimizations employed by the framework have a subtle impact on model convergence. Therefore, although the training time for a specific run might be reduced, the overall development time and complexity might increase.

Comments:

Adversarial training is slow, and the paper looks into how to accelerate AT through distributed training. The paper correctly identifies several techniques, such as large-batch, gradient compression, and FGSM, to improve the computation efficiency on single GPU as well as reducing the communication volume across GPUs and machines. The evaluation is thorough and convincing. The main concern is on the novelty side. All techniques are from existing work and have been studied heavily: FGSM reduces the amount of computation for inner maximization; gradient quantization reduces the communication volume; large-batch training improves GPU utilization. Therefore, the contribution is mainly on creating a framework that combines all the techniques and demonstrates empiricdally that they help scale out AT.

The reviewer appreciates the theoretical analysis of the convergence rate of DAT. However, similar convergence proofs have been given in each of the individual techniques in the past, as in [1-3]. Why isn't the convergence of DAT a simple composition of the three, or is there a more specialized aspect in the convergence proof for DAT?

Another main concern is the actual convergence impact of the composed framework. Each of the proposed techniques would have an impact on model convergence and is known to make training more difficult, e.g., each may lead to model divergence, slow convergence, or convergence to suboptimal local minima. How does DAT help avoid these challenges or does DAT actually impose more challenges because it involves the interaction of all three methods?

Question:

The evaluation does not include a comparison of Fast-AT under the 6x6 setting. Is it because there is no performance gain when training Fast-AT across multiple servers, or is it because Fast-AT does not support multi-node training? If it is the latter case, then it seems to be better to add Fast-AT (6x6) as a baseline since it does not seem to be too difficult to extend AT with DDP in PyTorch or multi-node training in TensorFlow, and it would show a better gap between the current multi-GPU multi-node training and DAT.

[1] Wang et. al. "On the Convergence and Robustness of Adversarial Training ", http://proceedings.mlr.press/v97/wang19i/wang19i.pdf

[2] You et. al., "Large Batch Optimization for Deep Learning: Training BERT in 76 minutes", https://arxiv.org/abs/1904.00962

[3] Alistarh et.al., "The Convergence of Sparsified Gradient Methods", https://papers.nips.cc/paper/7837-the-convergence-of-sparsified-gradient-methods.pdf

---

> ### Author Response · Authors · 2020-11-18
> **Response to Reviewer 1 - Part 3**
>
> ### **A comparison of Fast-AT under the 6x6 setting?**
>
> Thanks for the question. Fast-AT does not directly support the distributed training across distributed data over multiple computing nodes. Yes, it is possible to extend Fast-AT to a distributed version. In fact, such an extension becomes one of our baseline settings `DAT-FGSM w/o LALR except the use of cyclic learning rate in Fast-AT. However, we found that the conventional Fast-AT has a quite poor scalability versus the growth of data batch size (Appendix 4.1), leading to  a worse distributed baseline than DAT-FGSM w/o LALR. To further clarify this point, we exactly follow the ImageNet setting of Fast-AT (Wong et al., 2020), which consists of three training phases using different batch sizes (due to image resizing), $512$ for phase 1, $224$ for phase 2, and $128$ for phase 3. And we conduct **new experiments** by increasing the batch size to the maximum value that each training phase supports, given by $4096$ for phase 1, $1300$ for phase 2 and $384$ for phase 3. As a result, the direct distributed implementation of Fast-AT (Wong et al., 2020)  yields **43.67% (TA)** and **28.10% (RA)**, which are much worse than the performance of DAT-FGSM w/o LALR. Thus, compared to  the direct extension of Fast-AT (Wong et al., 2020), our considered DAT-FGSM w/o LALR is actually a stronger distributed baseline.
>
>
> We hope our response has addressed most of your concerns, and we hope it highlights our efforts in making a thorough study of distributed adversarial training when the centralized training becomes inapplicable. We are glad to continue discussion to address any other questions you may have.

---

> > ### Comment · AnonReviewer1 · 2020-11-22
> > **Comments**
> >
> > Thank you very much for the clarification. My concern on this issue has been addressed. It might be better to add this clarification in the paper with a few sentences.

---

> > > ### Author Response · Authors · 2020-11-24
> > > **Thank you very much**
> > >
> > > We are glad to learn that this concern has been addressed. Thanks again for your participation and insightful comments.

---

> ### Author Response · Authors · 2020-11-18
> **Response to Reviewer 1 - Part 2**
>
> ### **Overall development time and complexity might increase? Actual convergence impact of the composed framework?**
>
>  We do not think the overall development time and complexity will increase.
>
> First, although combined techniques (FGSM attack generation, randomized gradient quantization and layerwise adaptive learning rate) are used, we did not impose extra hyper-parameters that are difficult to tune. In fact, we follow the previously-established parameter setups except the use of cyclic learning rate in FGSM-based DAT. We found that such a learning rate trick does not benefit the large-batch setting as the number of iterations significantly decreases; see Appendix 4.1. Thus, we used the standard piecewise decay step size and an early-stop strategy suggested by [4].
>
> [4] Rice, Leslie, Eric Wong, and J. Zico Kolter. "Overfitting in adversarially robust deep learning." ICML, 2020.
>
> Second, none of FGSM attack generation, randomized gradient quantization and layerwise adaptive learning rate (LALR) is of high complexity and takes high computation time. By contrast, the computation gain from each aspect is apparent: FGSM simplifies inner maximization, gradient quantization reduces communication cost, and LALR supports the use of large data batches (and thus enables training over a small number of iterations). Their usage reduces the total training cost as verified in our implementations and experiment results. Also, each will not lead to divergence and slow convergence, proved by our theory. Yes, it may converge to suboptimal local optima, but we empirically show that the performance of a converged model (averaged over 3 runs with different random seeds per experiment; see the paragraph “evaluation setting” in Sec. 5) outperforms other distributed baselines.
>
>
> Third, we would like to remark that in practice, DAT is not able to achieve linear speed-up mainly because of the communication cost and other minor factors such as the rounding effect of batch size, and synchronizing cost across multiple machines. Let us take the ImageNet experiment in Table 1 as an example. When comparing the computation time of DAT-PGD with that of AT, the computation speed-up (by excluding the communication cost) is given by $(6022)/(1960-898) = 5.67$, consistent with the ideal computation gain using $6$x larger data batch size in DAT-PGD. In CIFAR-10 experiments, the rounding effect of batch size may also play a role in affecting the practical speed-up. Let $D $ be the size of the dataset, $b$ be the batch size per node, $p$ be the number of computing nodes, and $[\cdot]$ be the ceiling function. Then, the speed-up that can be achieved by DAT versus AT  is given by $\frac{ [D/b]}{ [D/(bp)] }$. In Table 1, when comparing  DAT-PGD with AT, we have $ [D/b] = 25$ and $ [D/(bp)] = 2$, and $\frac{ [D/b]}{ [D/(bp)] } = 12.5$. This is consistent with the practical computation speed-up measured by the computation time, $218/18.6 = 11.7$.
>
>
> I hope that our aforementioned response has mostly addressed the reviewer’s concerns. If any specific measurement on ‘overall development time and complexity’ can be suggested from the reviewer, we are glad to conduct further experiments for justification. At least, from our current results on model configuration, model convergence, communication and computation time, we did not see that our proposal leads to an increase in overall development time and complexity.

---

> > ### Comment · AnonReviewer1 · 2020-11-22
> > **Comments**
> >
> > Thank you very much for your detailed response! It is good that DAT does not introduce extra hyperparameters that are difficult to tune. However, my worry is that each of the techniques introduced (i.e., LALR, sparse gradients) speeds up training by introducing some extra hyperparameters that need to be tuned, and the complexity of the hyperparameter tuning grows combinatorially when combining all of them. For example, to speed up the training of a new model with a large batch, one often needs to tune the batch size, learning rates, LAMB coefficient (c_l, c_u). To leverage the spare gradient, one needs to tune the reduced bits and presumable when to introduce sparse gradients in order to avoid accuracy loss. If one does not get the desired accuracy using DAT, they have more hyperparameters that need to be tuned than using each technique alone, and the search space also becomes much larger, which increases the development complexity. To tune the hyperparameters, one needs to spend multiple trials on training, so it may end up taking an even longer time to train a model in order to get the desired accuracy. Furthermore, from the experiments, it seems some of the techniques cause accuracy degradation. For example, the quantized gradients cause non-trivial accuracy drop, e.g., 6.33% for DAT-FGSM on CIFAR-10, and RA drops by 5.22%. When seeing such an accuracy drop, it is unclear whether one should accept that as a trade-off in training speed and accuracy or should tune more hyperparameters to increase the accuracy. It would be helpful if the authors can clarify how DAT helps train a new model and deals with the potential combinatorial explosion issue in hyperparameters.

---

> > > ### Author Response · Authors · 2020-11-24
> > > **Further response**
> > >
> > >
> > > Thanks for your further comments and insightful explanation. In our current experiments (ResNet-18 and 50 over CIFAR and ImageNet), we did not intend to optimize the hyper-parameters (we chose their values following the commonly-used setup). However, we understood the reviewer's concern and agree that this may not a **universal** and **optimal** solution to the hyper-parameter configuration when training a new model. Spurred by that, we make further clarification and conduct additional experiments below.
> > >
> > > First, we decide not to tune the batch size since we have set a batch size as large as possible at each GPU.  Indeed, compared to the ImageNet setup of [Xie et al. 2019] ($32$ batch size per GPU over $128$ GPUs), we used the batch size $3072$ across $36$ GPUs, leading to a more aggressive **per-GPU utility**.
> > >
> > > Second, we did not tune the learning rate constant much because as (Figure 2; Rice et al., 2020) pointed out, the learning rate choice may not be a key factor when taking into account the effect of robust overfitting. We followed (Rice et al., 2020) and used the decay schedule. However, we agree with the reviewer that LAMB coefficients $(c_l, c_u)$ can further be tuned. Thus, we conducted an **additional experiment** to examine if the performance of DAT is sensitive to different choices of LAMB coefficients. In the new experiments, we fix $c_l = 0$ (we feel that this is a natural choice) but varies $c_u \in \{ 8, 9, 10, 11, 12 \}$ for *DAT-FGSM under CIFAR-10 using $18 \times 2048$ batch size*, where $c_u = 10$ is the default setup.  As we can see, both RA and TA are not quite sensitive to $c_u $ and the default choice yields the RA-best model (despite minor improvement).
> > >
> > > LAMB coefficient $c_u$ | TA  | RA
> > > ------------- | ------------- | -------------
> > >  $c_u = 8$  |   73.57    | 38.19
> > > $c_u = 9$  | 73.72 | 38.00
> > >  $c_u = 10$ (original choice)  | 73.42  | 38.55
> > > $c_u = 11$	  | 73.75  | 38.18
> > > $c_u = 12$	  | 73.63  |  37.87
> > >
> > > Third, we agree with the reviewer that when striking a balance with communication cost, the quantized gradients could cause a non-trivial accuracy drop in DAT. This motivates us to examine whether or not the issue comes from the gradient quantization itself or the combined hyper-parameter effect of DAT. Thus, we conduct an **additional experiment** by integrating a **centralized** method with gradient quantization operation on CIFAR-10 under the batch size $2048$ and $6 \times 2048$, respectively. We specify the centralized method as Fast AT with LALR, where LALR is introduced to improve the scalability of Fast AT to meet the batch size $6 \times 2048$. Due to the centralized implementation, we only need 1-sided gradient quantization (namely, no server-worker communication is involved). We report our results below.
> > >
> > > Centralized method | 8-bit quantization | Batch size  | TA  | RA
> > > ------------- | ------------- | ------------- | ------------- | -------------
> > > Fast AT  |  No |  2048  | 81.58    | 38.34
> > > Fast AT w. LALR  | Yes  | 2048 |  80.66 |  38.60
> > > Fast AT w. LALR | No | 6 x 2048 |  80.08 |  38.51
> > > Fast AT w. LALR | Yes| 6 x 2048 |  75.53 |  38.45
> > >
> > > As we can see, when the batch size $2048$ is used, Fast AT w. LALR performs as well as Fast AT even in the presence of 8-bit gradient quantization. On the other hand, as the larger batch size $6 \times 2048$ is used, Fast AT w. LALR can still preserve the performance when gradient quantization is *not* introduced. By contrast, Fast AT w. LALR in the presence of quantization encounters a 6.05% TA drop. This suggests that even in the non-DAT setting, 8-bit gradient quantization hurts the performance as the batch size becomes large. Thus, in DAT it is not surprising that 8-bit quantized gradients could cause a non-trivial accuracy drop, particularly for using 2-sided gradient quantization and a much larger data batch size ($\geq 18 \times 2048$ on CIFAR-10). The possible reason is that the quantization error cannot easily be mitigated as the number of iterations decreases (due to increased batch size under a fixed number of epochs). In our experiments, although we focused on the Ring-AllReduce communication configuration that calls for 1-sided (rather than 2-sided) gradient quantization, we agree with the reviewer that the number of quantization bits should be tuned as an aggressive choice might lead to performance degradation. It is also worth noting that such a phenomenon occurs in both centralized and distributed settings. This is a great comment! Thanks!
> > >
> > > Thanks again for the great suggestion, which motivates us to revisit and better understand the possible combinatorial issue in hyper-parameters. We will add the aforementioned discussion in the revised paper.

---

> ### Author Response · Authors · 2020-11-18
> **Response to Reviewer 1 - Part 1**
>
> We thank the reviewer very much for the insightful comments. We provide the detailed response below.
>
> ### **Limited technical contributions;  simple composition of [1-3]?**
>
> We are sorry to learn that our work is regarded as having limited contributions. Although either of the standalone techniques, adversarial training, gradient quantization, and large-batch standard training, have been established separately, justifying this seemingly straightforward combination **'actually works'** is **novel**. For example, in theory, DAT needs to quantify how descent errors from multiple sources (gradient estimation, quantization, adaptive learning rate, and imperfect inner maximization oracle) affect the convergence of DAT.  In fact, these standalone techniques for the first time are proven and shown effective when used jointly in DAT through our theoretical analysis. We strongly believe that it is highly non-trivial to develop a unified and theoretically-grounded DAT framework with extensive experiment support. We elaborate on our point below.
>
> **[Theoretical contribution]**  The convergence analysis of DAT is NOT a simple composition of [1-3]. By contrast, in DAT these individual techniques are coupled with each other and make our analysis far from trivial. We have highlighted our theoretical challenges in the first paragraph of the section “Convergence analysis of DAT ”. To the best of our knowledge, we are even not aware of any established convergence rate analysis for general large-batch min-max optimization problems. Here we make further clarifications:
>
>
> The fundamental challenge lies in how errors from multiple sources (gradient estimation, quantization, adaptive learning rate, and imperfect inner maximization oracle) are coupled and evolved with alternating updates in min-max optimization (AT). Different from [1-3], it is indeed challenging for us to quantify the descent of the objective value by taking into account these involved errors from multiple sources, and to derive the theoretical relationship between large data batch (across distributed machines) and the eventual convergence error of DAT. To be more specific, a new descent lemma (Lemma 2) is provided to measure the decrease of the objective value, which is used to deal with the nonlinear coupling between the multiple error sources and the true gradient. For outer minimization, the bias error term resulted from the layerwise normalization, (i.e., W, an upper bound of V, in the proof of Theorem 2) further contributes to the convergence rate in a square root of the oracle error (\varepsilon) along with stochastic gradient estimate and quantization error. We prove that under some mild assumptions the bias resulting from the layerwise normalization can still be compensated by increasing the batch size. For efficient communication, We are interested to learn any further feedback from R#1 on why our theoretical contribution is trivial to obtain from [1-3].
>
> **[Other contributions]** First, the proposed formulation and algorithm of DAT covers a broad range of distributed variants of adversarial training (AT), such as supervised AT, semi-supervised AT,   FGSM-based AT, and quantization-tolerant AT. This is the first thorough study on distributed robust training. Second, we have made a significant effort to evaluate the empirical performance of DAT in defense against PGD and C&W attacks, scalability across different computing configurations (AllReduce, parameter-server and HPC setups), advantage of using unlabeled data, and transferability of robustness from DAT pre-training to fine-tuning. We strongly believe that our work is much beyond just creating a framework that combines all the techniques.

---

### Official Review · AnonReviewer2 · 2020-10-29
**Clearer message on when to use this method**

**Rating:** 5
**Confidence:** 4

**Review:**

This paper proposed distributed adversarial training (DAT) for robust models. The method is a combination of PGD-like adversarial training, LARS-like large batch training, and quantizing gradients for communication efficiency in distributed training. The authors show convergence of adversarial training with LARS-like learning rate under layer-wise assumptions, and empirical results of DAT can scale to 6x6=36 GPUs and batch size 6*512 for ImageNet.

Pros

+ The paper is well written and easy to follow.
+ The convergence rate looks reasonable.
+ It is good to show that adversarial training can be accelerated through distributed settings.
+ Extensive experiments on CIFAR-10 and ImageNet.

Cons

- The general contribution of the paper is a bit incremental. The main message seems to be that LARS can help large-batch adversarial training?
- The convergence rate contribution seems incremental given the known adversarial training convergence proof and LARS-like convergence proof.
- As far as I know, scaling up learning rate (maybe with warmup) performs good for large batch training, we only need LARS for a certain regime, the ImageNet setting in table 1 where the batch size is 6*512 seems to fall in the regime where scaling LR works. Did the authors scale up LR for baselines without LARS?
- The Fast FGSM method (Wong et al. 2020) uses cyclic LR etc. to make convergence fast. Are those tricks applied here? Wong et al. reported CIFAR training in about 10 min, which is faster than the baseline in table 1 (52 sec/epoch * 100 epochs). I am worried the authors may target a less significant problem than we expected.
- [Xie et al. 2019 Feature Denoising for Improving Adversarial Robustness] and Kannan et al. 2018 use more GPUs than reported in the paper. And Xie et al. achieves better robustness on ImageNet. What is the practical advantage of this paper compared to Xie et al? What are the insights from the authors’ reimplementation of Xie’s method on CIFAR? ( DAT-LSGD in table 1)
- Why does communication time decrease when more GPU machines (24) are used?



==================== post rebuttal ==============================

I do not think my concerns are addressed by the discussion. However, I also think this is a well-written paper in general and I will not be upset if it is accepted.

My main concerns,
(1) The authors fail to show that the proposed method is non-trivial. I think this concern is raised by multiple reviewers. The authors keep clarifying the technical difficulty (especially the theory) of applying LARS etc, but my main concern is the necessity of these knobs added by authors. After a few rounds of discussion, we reach to a conclusion that adversarial training is different from standard training, but I do not think that could be considered insights from this paper. I would strongly suggest authors consider explaining why LARS is necessary by either theory or intuitive insights, and make it clear what exactly the difference is.
(2) The authors claim contributions for large scale setting (ImageNet with large number of available GPUs), but the experiments are somewhat worse than previous results. Lacking computation resources is a good excuse, but since the authors claimed they can use larger batch size with smaller number of GPUs, I do not see a technical reason why they cannot use their method to re-run the large scale experiments to directly compare with previous results.

---

> ### Author Response · Authors · 2020-11-18
> **Response to Reviewer 2 - Part 2**
>
> ### **Cyclic LR applied? Less significant problem than we expected?**
>
> Thanks for raising this question and sorry for the confusion. We have made detailed discussion on the use of cyclic LR (Wong et al. 2020) in the paragraph at the end of page 6 and Appendix 4.1.
>
> * Yes, the use of a cyclic learning rate trick can further accelerate the Fast AT algorithm in (Wong et al., 2020). However, such a trick does not help the large-batch setting as the number of iterations significantly decreases; see Appendix 4.1. It was also recently shown in [1] that the sensitivity of AT to LR can be mitigated by an early-stop strategy, leading to a more principled min-max robust training framework. Thus, we decided to use the standard piecewise decay step size and an early-stop strategy suggested by [1] in our baseline setup.
>
> * We understood the reviewer’s concern on “Wong et al. reported CIFAR training in about 10 min, which is faster than the baseline in Table 1 ($52$ sec/epoch $\times 100$ epochs).” This is also the reason for us to study the scalability of the proposed DAT versus the very large-batch size ($24 \times 2048$) under CIFAR-10 in Table 1. As we can see, our approach took ($5$ sec/epoch $\times 100$ epochs) computation time less than 10 minutes. Thus, we believe that the acceleration achieved by DAT is promising as it is independent of cyclic LR heuristics.
>
> [1] Rice, Leslie, Eric Wong, and J. Zico Kolter. "Overfitting in adversarially robust deep learning." arXiv preprint arXiv:2002.11569 (2020)
>
> ### **Practical advantage of this paper compared to Xie et al? What are the insights from the authors’ reimplementation of Xie’s method on CIFAR? ( DAT-LSGD in table 1)**
>
> Thanks for the question. We do *not* think that it is fair to directly compare our proposed DAT with [Xie et al. 2019], since the latter used different model architectures by incorporating feature denoising. In contrast, DAT does not rely on architecture modification. Thus, to enable a fair comparison, we use the same training recipe (large-batch SGD) as [Xie et al. 2019] in the DAT setting, leading to the baseline method DAT-LSGD. We believe that a principled DAT approach should be dataset-agnostic. Thus, we begin by conducting CIFAR-10 experiments for both our approach and Xie’s method DAT-LSGD. Moreover, the CIFAR-10 experiments enable us to investigate the scalability of DAT versus a wide range of batch size settings under our current resource budget. As shown in Figure 2, our approach scales more gracefully than the baseline (without losing much performance as the batch size increases along with the number of computing nodes). This is a practical advantage. To further confirm our advantage, we conduct **additional experiments** by comparing the proposed DAT-FGSM with DAT-LSGD on **ImageNet** across multiple machine-GPU configurations. The **updated Figure 1** demonstrates that our approach outperforms DAT-LSGD under not only CIFAR-10 but also ImageNet.
>
> Due to the limitation of our computing resources, we are not able to use as many GPUs as Xie et al. 2019 and Kannan et al. 2018. However, our current results under various computing configurations and adversarial scenarios have clearly demonstrated the advantages of our proposal in generalizability (supervision and semi-supervision, pre-training and fine-tuning, PGD and FGSM attack generation) and scalability (large-batch learning and quantization-aware communication). For example, our used batch size $3072$ ($6 \times 512$, across $36$ GPUs) on ImageNet is really a large-batch setting compared to many existing adversarial training implementations. Although  [Xie et al. 2019] used batch size $4096$,  it had access to $128$ GPUs, yielding $32$ batch size per GPU that is smaller than our case. We hope that the reviewer will not down-grade our contributions due to our actual GPU resource limitation, which happened in many academic and industrial institutions.
>
> ### **Why does communication time decrease when more GPU machines ($24$) are used?**
>
>  In Table 1, the per-epoch communication time decreases when more GPU machines ($24$) are used, since a larger batch size allows a smaller number of iterations per epoch, leading to fewer communication times per epoch among machines. As a result, the total communication cost per epoch, given by the number of iterations multiplying the communication cost per iteration, decreases.
>
>
> We hope our response has addressed most of your concerns, and we hope it highlights our efforts in making a thorough study of distributed adversarial training. We are glad to continue a discussion to address any other questions you may have.

---

> > ### Comment · AnonReviewer2 · 2020-11-24
> > **Clear message on when to use this method**
> >
> > I thank the authors for the detailed response. I do not want to be picky here, but my main point here is to get a clear message on when researchers should consider the proposed method.
> >
> > (1) First of all, I consider this paper an empirical research, not a theoretical research. I acknowledge the potential difficulty of combining LARS, gradient quantization and adversarial training. However, each piece of the analysis has shown in previous works. The complexity of combining these analysis would become a none problem if people simply use distributed setting similar to small batch setting, like [Xie et al. 2019] .
> >
> > (2) Let me be more specific on my question on the necessity of LARS. In [Goyal et al. 2018 Accurate, Large Minibatch SGD: Training ImageNet in 1 Hour], it is OK to scale up batch size to 8k without LARS for standard large batch training. The authors in this paper use a batch size of 6*512 ~ 3k, which could be trained without LARS. Why LARS is necessary for adversarial training? It is interesting if adversarial training is different from standard training, but the fact the authors did not directly answer my question worries me.
> >
> > (3) For the comparison with [Xie et al. 2019] and [Wong et al. 2020]. Again, do not want to be picky here. What I am trying to figure out is when researchers should consider the proposed method.
> > (3a) The proposed method could be slightly faster than [Wong et al. 2020]. But is 600 secs with 24 GPUs (this paper) more interesting than 500 secs with 1 GPU (in [Wong et al. 2020])?
> > (3b) I understand it is hard to request 128 GPUs to compare with [Xie et al. 2019], and it is not necessary! My main concern is that this paper extensively claimed contributions for a large scale setting, but most of the experiments are (relatively) small scale. It looks to me the take away message is that for very limited resources (4 GPUs on a single workstation), researchers should consider Free [Shafahi et al. 2019] or  [Wong et al. 2020]; for rich resources (more than 128 GPUs), researchers should consider [Xie et al. 2019] or [Qin et al. 2019]. Only when we have resources some where in between (maybe 36 GPUs like in this paper), we should consider the proposed method with LARS.
> > (3c) I do not understand the authors argument on per GPU batch size. Why does it matter? Please clarify.

---

> > > ### Author Response · Authors · 2020-11-24
> > > **Thank you and further response**
> > >
> > > Thanks for your further comments. We provide a pointwise response below.
> > >
> > > **(1)** Compared to [Xie et al. 2019], our method would be preferred for accelerating adversarial training over distributed machines. In the revised Figure 1, we have shown that our proposal outperforms [Xie et al. 2019] in both CIFAR-10 and ImageNet.
> > >
> > > **(2)** Thanks, your specific comment makes us better understand your question. As we have shown in Table 1, under the 3k batch size, training without LALR will lead to significant performance degradation. We agree with the reviewer that "it is OK to scale up batch size to 8k without LARS for standard large batch training". However, adversarial training, in the form of min-max optimization, makes the case much more involved than the standard training, which is a min-only problem. The two-layer (min-max) game makes the defender and the attacker co-evolved, and the batch size tolerance of adversarial training could be different from standard training. Intuitively, LAMB resorts to the layer-wise adaptive learning rate and momentum. Thus, the used gradient normalization and gradient averaging in LAMB can help reduce the variance of the evolved descent errors given a reduced number of iterations (when the large batch size is used).
> > >
> > > **(3a)** Compared to [Wong et al. 2020], DAT can be used for private data allocated across multiple machines. And as we presented in Table 2, it is beneficial to incorporate unlabeled data. Thus, as a large amount of unlabeled data are used, it is also nice to use DAT to extend the 1-GPU case.
> > >
> > > **(3b)** The 36 GPU limitation is due to our resource budget. Sorry, this is not our take-away message "Only when we have resources some where in between (maybe 36 GPUs like in this paper), we should consider the proposed method with LARS."
> > > Our main message is that DAT is a principled method to scale up adversarial training via distributed learning when data are distributed across multiple machines and GPU resources are limited to each machine. In spite of our resource budget constraint, we tried the best to compare our method with the baseline methods when the batch size is set as large as possible under 36 GPUs.
> > >
> > > **(3c)** Sorry for the confusion. The per GPU batch size was highlighted because we have purposefully set the batch size to meet the maximum utility of each GPU under our resource constraint.

---

> ### Author Response · Authors · 2020-11-18
> **Response to Reviewer 2 - Part 1**
>
> We thank Reviewer 2 very much for the detailed summary of our paper and the constructive comments. Our detailed response is listed below.
>
> ### **Incremental general contribution; the Main message is that LARS can help large-batch adversarial training?**
>
> It is not incremental to develop a unified and theoretically-grounded distributed adversarial training (DAT) framework. Yes, one of our main messages is that layerwise adaptive learning rate (LALR) can help large-batch adversarial training. However, justifying such a conclusion is far from trivial in both theory and practice:
>
> *  In theory, DAT needs to quantify how descent errors from multiple sources (gradient estimation, quantization, adaptive learning rate, and imperfect inner maximization oracle) affect the convergence of DAT. In particular, one key challenge is the nonlinear coupling between the biased gradient estimate resulting from LALR and the additional error generated from alternating updates in DAT. Our convergence rate analysis (Theorem 1) is not just a simple combination of LARS-like convergence proof and adversarial training convergence proof that are designed for centralized optimization (see more details in our next response A2)
>
> * In practice, we have made extensive efforts to show how LALR excellently performs in DAT. For example, the performance improvement of our approach over the large-batch SGD (LSGD) baseline (Xie et al. 2019) and directly distributed implementations of AT or Fast-AT have been shown in Figure 1, Table 1, and Appendix 4.1 under different distributed learning settings. We have also shown that the scalability of DAT lies in not only the large-batch computation setting but also the pre-training + fine-tuning setting (Figure 3) and the quantization-aware communication setting (Appendix 4.6).
>
> We believe that our formulation, algorithm, theory, and experiments cover a broad range of distributed versions of AT variants, such as supervised AT, semi-supervised AT,  FGSM-based AT, and quantization-tolerant AT. This is the first thorough study of distributed robust training.
>
> ### **Convergence rate contribution seems incremental given the known adversarial training convergence proof and LARS-like convergence proof?**
>
> The incorporation of layerwise adaptive learning rate (LALR) makes the convergence rate analysis of DAT far from trivial. Besides, the error induced by gradient quantization imposes extra difficulties. The fundamental challenge lies in how errors from multiple sources (gradient estimation, quantization, adaptive learning rate, and imperfect inner maximization oracle) are coupled and evolved with alternating updates in min-max optimization (AT). Different from existing adversarial training convergence proof (without considering gradient quantization and adaptive learning rate) and LARS-like convergence proof (without considering gradient quantization and min-max optimization), it is indeed challenging for us to quantify the descent of the objective value by taking into account these involved errors from multiple sources, and to derive the theoretical relationship between large data batch (across distributed machines) and the eventual convergence error of DAT. To be more specific, a new descent lemma (Lemma 2) is provided to measure the decrease of the objective value, which is used to deal with the nonlinear coupling between the multiple error sources and the true gradient. For outer minimization, the bias error term resulted from the layerwise normalization, (i.e., W, an upper bound of V, in the proof of Theorem 2) further contributes to the convergence rate in a square root of the oracle error ($\varepsilon$) along with stochastic gradient estimate and quantization error. We prove that under some mild assumptions the bias resulting from the layerwise normalization can still be compensated by increasing the batch size in min-max alternative optimization.
>
> ### **Did the authors scale up LR for baselines without LARS?**
>
> Thanks for the question. Yes, scaling up the learning rate (with warm-up) is useful for large-batch training. For example, this was used in large-batch SGD (Xie et al., 2019), which we call  DAT-LSGD. We also used this strategy for DAT with and without LALR: a 5-epoch warm-up with linearly increased learning rate up to 0.01; see training details in Appendix 4.2

---

### Author Response · Authors · 2020-11-24
**Summary of response**

Dear Reviewers,

Thank you very much for the efficient handling of our manuscript. As the reviewer-author discussion phase will end soon, we would like to summarize the highlights of our response below and invite reviewers to check our pointwise response for details.

*Contribution*: Technically, we have made detailed clarification on why our contribution is not incremental. To the best of our knowledge, we are even not aware of any established convergence rate analysis for large-batch min-max optimization prior to our work. Practically, it is also not incremental to conduct careful empirical studies of DAT in generalizability (supervision and semi-supervision, pre-training and fine-tuning, PGD and FGSM attack generation) and scalability (large-batch learning and quantization-aware communication). Thus, we do believe that our contribution lies in both theory and practice.

*Baseline comparison*: We have made detailed clarification and conducted additional experiments on a) why distributed methods are needed vs. centralized methods, b) why conventional fast AT does not directly support distributed implementation, c) why the cyclic learning rate is not proper for DAT, d)  comparison with more centralized AT or Fast AT variants and e) comparison with DAT-LSGD on ImageNet.

Based on our response, we have also updated our manuscript with major modifications marked by blue color. We hope that our rebuttal and the revised paper have mostly addressed reviewers' concerns and our effort would properly be recognized.

Thanks a lot,

Authors,

---

### Decision · Program_Chairs · 2021-01-07
**Final Decision**

**Decision:**

Reject

**Comment:**

In this paper, the authors claim to propose a distributed large-batch adversarial training framework to robustify DNN.
Although the authors made efforts to clarify reviewers' concerns,  it is clear that the authors still cannot convince some reviewers in several points after several rounds of discussion between reviewers and authors.

The reviewers were not in consensus on acceptance and some concerns were still not clearly addressed in the rebuttal phase.
Hence, I recommend acceptance only if there is a room.